# EMERGENCE OF EXPLORATION IN POLICY GRADIENT REINFORCEMENT LEARNING VIA RETRYING

## ABSTRACT

In reinforcement learning (RL), agents benefit from exploration because they repeatedly encounter the same or similar states, where trying different actions can improve performance or reduce uncertainty; otherwise, a greedy policy would be optimal. We formalize this intuition with **ReMax**, an objective that evaluates a policy by the expected maximum return over $M$ samples ($M \in \mathbb{N}$), while accounting for return uncertainty. Optimizing this objective induces stochastic exploration as an emergent property, without explicit bonus terms. For efficient policy optimization, we derive a new policy-gradient formulation for ReMax and introduce **Re**Max **PPO** (**RePPO**), a PPO variant that optimizes ReMax while generalizing the discrete retry count $M$ to a continuous parameter $m > 0$, enabling fine-grained control of exploration. Empirically, RePPO promotes exploration without bonuses on the MinAtar and Craftax benchmarks.

## 1 INTRODUCTION

Exploration is a central problem in reinforcement learning (RL) and has been extensively studied (Sutton & Barto, 2018; Haarnoja et al., 2018; Ziebart et al., 2008; Mnih et al., 2016). A prevailing approach adds bonuses, such as entropy (Haarnoja et al., 2018; Ziebart et al., 2008) or count-based bonuses (Bellemare et al., 2016; Ostrovski et al., 2017), to the environment's reward to encourage exploration explicitly. Other lines of work instantiate posterior sampling (Thompson, 1933) using ensembles or Bayesian networks (Osband et al., 2016; 2018; 2019). We study a distinct mechanism that drives exploration via greedy reward maximization.

The objective of RL is not to maximize the reward on the current trial, but rather to learn to maximize the rewards after several trials. Exploration matters because it enables achieving higher rewards on subsequent trials. Without the opportunity for retrying, exploration is unnecessary: the rational choice is the action currently believed to yield the highest reward. Environmental uncertainty also motivates exploration, encouraging attempts at alternative actions for information gathering. If there is no uncertainty, we do not need to explore: the problem is reduced to pure optimization.

Building on this principle—*decision-makers retry under uncertainty*—we propose the **ReMax** objective, which formalizes exploration as reward maximization under uncertainty. We first present ReMax in a bandit setting and compare it to the standard RL objective.

**The ReMax objective (bandit).** Let $K \geq 2$ be the number of actions, let $\mu = (\mu_1, \ldots, \mu_K)$ denote per-action values, and let $\Pi$ be a distribution over $\mu$. For a policy $\pi \in \Delta^{K-1}$ ($\Delta^{K-1}$ is the probability simplex over $K$ actions) and $M \in \mathbb{N}$,

$$J_{\mathrm{RL}}(\pi) := \mathbb{E}_{A \sim \pi}\left[\mu_A\right], \qquad J_{\mathrm{ReMax}}^M(\pi) := \mathbb{E}_{\mu \sim \Pi}\left[\mathbb{E}_{A_{1:M} \sim \pi}\left[\max_{1 \leq m \leq M} \mu_{A_m} \mid \mu\right]\right]. \tag{1}$$

We highlight the difference between the two objectives by coloring terms in red and blue. The blue component, the expected maximum over $M$ draws, represents the spirit of **retrying**, where we *score the policy by the best outcome across multiple trials*. With $M = 1$ and fixed rewards, this reduces to the standard RL objective $J_{\mathrm{RL}}(\pi)$, and the optimal policy is deterministic (Sutton & Barto, 2018). However, for $M \geq 2$, depending on the reward distribution, a stochastic policy can be optimal (Sec. 2). On the other hand, the red component, which encodes (epistemic) **uncertainty** (Ghosh et al., 2021), stems from the limited knowledge of the environment and varies over the course of

the agent's exploration. We address the uncertainty by explicitly modeling the distribution (Osband et al., 2016) or through samples from a nonstationary estimator of returns (Moalla et al., 2024).

Some prior works have introduced curiosity-based bonuses to encourage exploration, including pseudo-count methods (Bellemare et al., 2016; Lobel et al., 2023) and prediction-error-based approaches (Pathak et al., 2017; Burda et al., 2019). While these methods have proven effective in ALE game domains, they typically rely on additional models to estimate the statistics required for constructing the bonuses. This reliance not only increases algorithmic complexity but also imposes extra computational overhead.

Unlike methods that add explicit bonuses, ReMax induces exploration *without* bonuses by optimizing a purely reward-based objective. Recent work in LLM training for reasoning tasks has studied retry-style objectives such as pass@$K$ (Walder & Karkhanis, 2025; Tang et al., 2025), which share a similar idea to ours. The crucial distinction is that ReMax explicitly accounts for reward uncertainty and nonstationarity, whereas LLM reasoning tasks typically assume fixed, verifiable rewards. For a broader discussion and connections to related work, see App. A.

Throughout this paper, we address the following question:

> Can we promote exploration without adding explicit bonuses by optimizing ReMax in RL?

To answer this question, we organize the paper as follows.

**Step 1: Empirical study in Bandits.** We first empirically illustrate the core idea of how ReMax (defined in Eq. (1)) induces effective exploration in bandits in Sec. 2. We show that as the retry parameter $M$ increases, the optimal policy becomes more exploratory. Moreover, ReMax adapts the degree of exploration to the scale of reward uncertainty and, in a posterior bandit setting, exhibits sublinear regret empirically.

**Step 2: ReMax in RL** We then define the ReMax objective for RL in Sec. 3. Unlike bandits, RL involves state transitions, which hinder retrying multiple actions from the same state to observe their returns, even with a simulator. We therefore emulate such retries via queries to a $Q$-function and discuss possible instantiations of ReMax in RL.

**Step 3: Policy Gradient for ReMax.** To optimize ReMax, we develop a practical policy-gradient (PG) method in Sec. 4. We derive a new PG formulation that is estimable from trajectory returns and generalize the integer draw count $M$ to a positive real parameter $m > 0$, enabling finer control of the exploration–exploitation trade-off. Building on this formulation, we introduce **ReMax PPO** (RePPO), a PPO-based deep actor–critic algorithm (Schulman et al., 2017).

**Step 4: Experiments.** Finally, we evaluate RePPO on MinAtar (Young & Tian, 2019) in Sec. 5. RePPO optimizes ReMax without any exploration bonuses, achieves better performance, and maintains higher policy entropy than PPO with an entropy bonus. We also find that the continuous retry parameter provides practical flexibility, with peak performance around $m = 1.2$–$1.4$. We also evaluated RePPO on Craftax (Matthews et al., 2024), a larger-scale open-ended RL environment, where the agents need to explore and adapt to the environment over the training. We found that RePPO achieves competitive performance compared to a tuned entropy-regularized PPO, despite not using the exploration bonus. Overall, ReMax emerges as a promising objective for exploration in RL.

## 2 ReMax in Bandits: An Empirical Study

The goal of this section is to provide an intuitive understanding of how ReMax (defined in Eq. (1)) promotes exploration through its objective design (retrying and uncertainty). In Sec. 2.1, we illustrate several fundamental properties of ReMax by explicitly designing a distribution over the reward $\Pi$. In Sec. 2.2, we consider a more realistic setting where the distribution is estimated from data and validate the effectiveness of ReMax for exploration.

### 2.1 Warm-up: Properties of ReMax.

**ReMax optima yield stochastic policies.** This example illustrates how the retry mechanism induces stochastic behavior. Consider a two-armed bandit with $\mu = (0, 1)$ w.p. 0.75 and $\mu = (1, 0)$

w.p. 0.25. For the RL objective, the optimal policy is deterministic and always chooses arm 1. In contrast, for the ReMax objective in Eq. (1), the optimal policy is stochastic. The reason is that repeatedly pulling the same arm does not increase the ReMax value, whereas pulling both arms ensures that the rewarding arm is chosen (since exactly one arm yields reward 1). Since $\mu = (0, 1)$ is more likely, a limited retry budget $M$ requires assigning sufficient mass to arm 1 to avoid missing it.

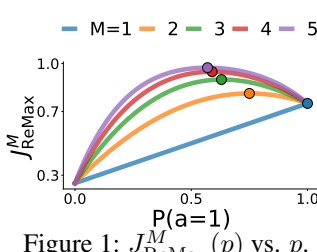

Figure 1: $J_{\text{ReMax}}^M(p)$ vs. $p$.

As $M$ grows, however, the policy can afford to explore arm 0 more often, and the optimal distribution becomes increasingly exploratory. Fig. 1 confirms this intuition by plotting the ReMax objective $J_{\text{ReMax}}^M(p)$ (vertical axis) against $p := \pi(a{=}1)$ (horizontal axis) for $M = 1, 2, 3, 4, 5$. The ReMax value is analytically calculated (see App. C.1). For each $M$, the optimal probability $p^*$ is indicated by a dot. As expected, when $M = 1$, the optimal policy is deterministic with $p^* = 1$ and value 0.75; for $M \geq 2$, the optimal policy gets more exploratory as $M$ grows, and the optimal value approaches 1. As such, the retry mechanism induces stochastic behavior in the presence of uncertainty.

**ReMax adapts exploration to reward uncertainty.** The previous example showed that ReMax induces stochastic behavior through the retry mechanism, allowing agents to try the less rewarding arm. Through the following example, we show that ReMax does not merely produce stochastic behavior but rather adapts to the magnitude of reward uncertainty. To verify this, we consider a two-armed Bernoulli bandit where the reward distribution $\Pi$ is as follows: $\mu_i = \alpha_i X_i$, where $X_i \sim \text{Bernoulli}(p_i)$ and $\mathbb{E}[\mu_i] = \alpha_i p_i$. We fix $p_0 = 1$ and $\alpha_0 = 2$, and vary $\alpha_1$ from 1 to 10, adjusting $p_1$ so that $\alpha_1 p_1 = 1$ remains constant, so that the *variance* varies while the *mean* remains fixed. Fig. 2 shows the

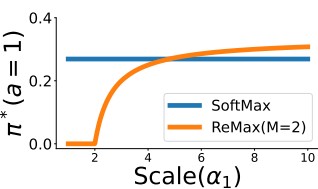

Figure 2: ReMax vs. Softmax.

optimal probability of selecting arm 1, $\pi^*(a = 1)$, for $M = 2$, alongside that of the softmax policy, which is the analytical optimum of the RL objective with an entropy bonus. Details of the settings appear in App. C.2. When $\alpha_1 \leq 2$, arm 1 is never chosen since its maximum cannot exceed that of arm 0. As $\alpha_1$ increases beyond 2, ReMax increasingly favors arm 1, reflecting its adaptiveness to rare but high-reward outcomes, potentially yielding effective exploration. In contrast, Softmax remains flat across $\alpha_1 \in [1, 10]$ because it depends solely on the mean, which is fixed.

**ReMax with deterministic rewards.** The previous examples focused on ReMax with stochastic rewards. We now turn to the deterministic setting and show that, *even with fixed rewards*, ReMax reshapes the policy gradient and thus modulates convergence by changing the retry parameter.

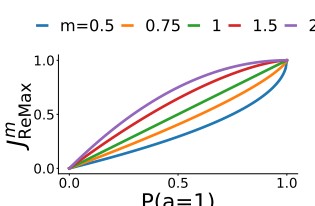

Figure 3: $J_{\text{ReMax}}^m(p)$ vs. $p$.

Consider a binary bandit with rewards fixed to $(0, 1)$ and let $p := \pi(a{=}1) \in [0, 1]$. In this case, the ReMax objective for $M$ retries admits a closed form; we have, for $m > 0$, $J_{\text{ReMax}}^m(p) = 1 - (1 - p)^m$, where $m$ can be an integer or a positive real. Fig. 3 plots $J_{\text{ReMax}}^m(p)$ for $m = 0.5, 0.75, 1, 1.5, 2$, where the inclusion of non-integer $m$ prepares for our generalization of ReMax to continuous $m$ in Sec. 3 and Sec. 4. The maximizer remains $p^* = 1$ for all $m$, indicating that after the agent has explored the environment enough to remove epistemic uncertainty, the policy will eventually converge to the optimal policy. However, the local geometry near high $p$ depends strongly on $m$: larger $m$ flattens the objective and reduces gradient magnitudes, whereas smaller $m$ sharpens the curvature and amplifies gradients. Thus, tuning $m$ controls convergence even in deterministic settings: $m > 1$ slows updates (encouraging exploration), while $m < 1$ accelerates them, mitigating the slow convergence often observed with softmax policies (Hennes et al., 2019). Furthermore, the curves for non-integer $m$ naturally fit within those with integer $m$ (e.g., $m = 1.5$ fits between $m = 1$ and $m = 2$), suggesting finer-grained control over policy convergence speed via the continuous parameter $m$.

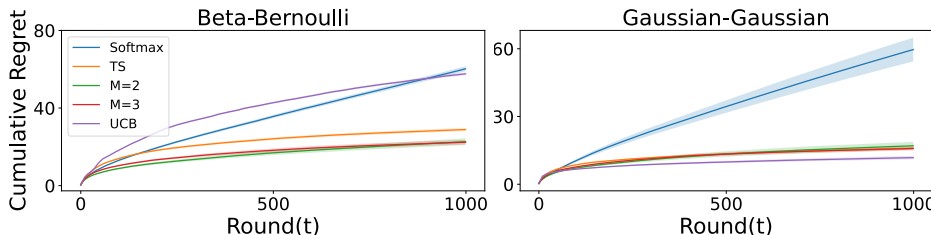

Figure 4: Average cumulative regret and standard error over 256 runs. ReMax with $M = 2$ and $M = 3$ achieves sublinear cumulative regret comparable to UCB and Thompson sampling.

## 2.2 BANDIT WITH POSTERIOR: EMPIRICAL SUBLINEAR REGRET.

In the previous section, we intentionally designed the distribution over reward $\Pi$ to illustrate properties of ReMax. However, in practice, the distribution is *estimated*, and updated from the observed data as the agent explores the environment. To validate the effectiveness of ReMax in such a realistic setting, we consider a $K$-armed bandit with posterior updates and optimize ReMax using samples from the evolving posterior (Thompson, 1933). We evaluate its cumulative regret, confirming the effectiveness of ReMax for exploration.

**Problem setup.** At the beginning of each run, the means $(\mu_1, \ldots, \mu_K)$ are drawn from a ground-truth distribution $\Pi^*$ and fixed for the run, with $\mu^* = \max_i \mu_i$. At round $t \in [T]$, we choose $A_t$, observe $R_t$ with mean $\mu_{A_t}$, and update $\Pi_{t+1}$ from $\Pi_t$ using a Bayesian update. We assume that the prior is $\Pi_0 = \Pi^*$. ReMax optimizes $\widehat{J}^M_{\text{ReMax}}(\pi_\theta)$ using samples from $\Pi_t$ to produce $\pi_t$, from which an action is sampled; see App. C.3 for details. We study two settings: (i) **Beta–Bernoulli**, with prior $\Pi_0$ being $\text{Beta}(1, 1)$ for all arms and $R_t \sim \text{Bernoulli}(\mu_{A_t})$; and (ii) **Gaussian–Gaussian**, with prior $\Pi_0$ being $\mathcal{N}(0, 1)$ for all arms and $R_t \sim \mathcal{N}(\mu_{A_t}, 1)$. We compare our approach with Thompson sampling (Thompson, 1933; Agrawal & Goyal, 2017; Honda & Takemura, 2014) and UCB (Auer et al., 2002), both of which have sublinear regret guarantees. To compare with entropy-regularized exploration, we prepared a Softmax baseline, where we took the softmax of the posterior means and selected the arm following the distribution. We set $K = 10$, $T = 1000$, and $M \in \{2, 3\}$ for ReMax. Instantaneous regret is defined as $\mu^* - \mu_{A_t}$, and we report the mean $\pm$ standard error of cumulative regret over 256 runs. Full experimental details are in App. C.3.

**Results.** ReMax exhibits empirically sublinear cumulative regret, comparable to UCB and Thompson sampling, while Softmax incurs higher cumulative regret, especially for the Gaussian-Gaussian bandit (Fig. 4). Our aim is not to claim superiority but rather to demonstrate that ReMax yields effective exploration in practice. We do not yet provide a theoretical proof of an upper bound on the cumulative regret, leaving it for future work.

## 3 REMAX IN RL

Having established the effectiveness of ReMax for exploration in bandit settings, we now extend it to RL. We model RL as an episodic Markov decision process (MDP) (Puterman, 2014) $\mathcal{M} = (\mathcal{S}, \mathcal{A}, r, P, T)$ with *discrete* actions $\mathcal{A} = \{1, \ldots, K\}$, reward $r : \mathcal{S} \times \mathcal{A} \rightarrow \mathbb{R}$, transition kernel $P : \mathcal{S} \times \mathcal{A} \times \mathcal{S} \rightarrow [0, 1]$, and horizon $T \in \mathbb{N}$. Given a policy $\pi : \mathcal{S} \times \mathcal{A} \rightarrow [0, 1]$ and a trajectory $\tau = (s_0, a_0, \ldots, s_T)$ induced by $(\pi, P)$, the objective is to maximize the expected return $\mathbb{E}_{\tau \sim (\pi, P)}[\mathcal{R}(\tau)]$ with $\mathcal{R}(\tau) := \sum_{t=0}^{T-1} r(s_t, a_t)$. When emphasizing the environment, we subscript the notation with $\mathcal{M}$, such as $r_{\mathcal{M}}(s, a)$ and $\mathcal{R}_{\mathcal{M}}(\tau)$.

In extending ReMax from bandits to RL, two issues arise: the unit of **uncertainty** and the feasibility of **retries**. Uncertainty in RL concerns the entire environment, including rewards and transitions, so we place a distribution over MDPs, $\mathcal{M} \sim P(\mathcal{M})$, capturing (epistemic) uncertainty over unexplored regions of the environment (Ghosh et al., 2021). Retry in RL would naively require multiple rollouts from the same state until termination, which is infeasible without a resettable simulator (Ecoffet et al., 2021) and often prohibitively expensive even with one. To address this, we introduce function

approximation via a Q-function $Q_{\mathcal{M}}^{\pi}(s,a) := \mathbb{E}_{\tau \sim (\pi,P)}[\mathcal{R}_{\mathcal{M}}(\tau) \mid s,a]$, replacing Monte Carlo returns from $(s,a)$ with queries to $Q_{\mathcal{M}}^{\pi}$. To make the retry structure explicit, we fix a state $s \in \mathcal{S}$ at which retries are considered. The ReMax objective is defined as

$$J_{\text{ReMax}}^M(\pi, s) := \mathbb{E}_{\mathcal{M} \sim P(\mathcal{M})} \left[ \mathbb{E}_{A_{1:M} \sim \pi} \left[ \max_{1 \leq m \leq M} Q_{\mathcal{M}}^{\pi}(s, A_m) \right] \right]. \tag{2}$$

As in the bandit setting, setting $M = 1$ with a fixed environment $\mathcal{M}$ recovers the standard RL objective, and the optimal policy is deterministic (Sutton & Barto, 2018). In Eq. (2), all within-environment randomness is already absorbed into $Q_{\mathcal{M}}^{\pi}$, so the dependence on $\mathcal{M}$ is only through $Q_{\mathcal{M}}^{\pi}$. Let $\mathcal{Q}$ denote the induced distribution over Q-functions; replacing the outer expectation over $\mathcal{M}$ by one over $\mathcal{Q}$ yields a practical form for optimization:

**Definition 1** (ReMax objective with Q distribution)**.**

$$J_{\text{ReMax}}^M(\pi, s, \mathcal{Q}) := \mathbb{E}_{Q \sim \mathcal{Q}} \left[ \mathbb{E}_{A_{1:M} \sim \pi} \left[ \max_{1 \leq m \leq M} Q(s, A_m) \right] \right]. \tag{3}$$

For a fixed state, the Q-values $Q(s, \cdot)$ form a $K$-dimensional vector, reducing the problem to the bandit case. Therefore, we can expect that ReMax enjoys the exploration advantages observed in Sec. 2. In practice, optimizing the ReMax objective involves several algorithmic considerations.

**Modeling Q-function uncertainty.** The objective in Eq. (3) can be estimated given samples of Q-values from the distribution $\mathcal{Q}$, leading to the following two approaches: (1) **Explicit modeling:** The obvious approach is to explicitly model $\mathcal{Q}$ using ensemble methods or Bayesian approaches (Osband et al., 2016; 2018; 2019; An et al., 2021). Model-based methods (Hafner et al., 2019) likewise capture environment-level uncertainty and induce the corresponding Q-distribution. (2) **Implicit modeling:** It is well known that in standard deep RL, Q-estimates by neural networks $Q_{\phi}(s, a)$ are inherently nonstationary due to distribution shift in the collected data and algorithmic randomness (Moalla et al., 2024; Tang & Berseth, 2024). Therefore, this nonstationarity forms an *implicit* distribution over the Q-function, and, for each update, we treat the predicted Q-values at state $s$, $Q_{\phi}(s, \cdot)$, as a sample from the distribution. As in the bandit setting (Sec. 2), ReMax is expected to adapt to this variability for exploration.

**Computing the expected maximum over $M$ trials.** Suppose a sample of Q-values for a state $s$ is given by $q = (Q(s, a_1), \ldots, Q(s, a_K))$ with $Q \sim \mathcal{Q}$ via either approach in the previous paragraph. The question is how to compute the inner expectation for a fixed $q$, defined as $J_{\text{ReMax}}^M(\pi, s, q) := \mathbb{E}_{A_{1:M} \sim \pi} [\max_{1 \leq m \leq M} q_{A_m}]$. One straightforward approach is to sample $M$ actions from the policy $\pi$ and compute the maximum, which yields an unbiased estimator but incurs high variance. Instead, we provide a closed-form expression for the inner expectation as follows.

**Proposition 1** (Closed-form expression of the inner expectation)**.** Given a Q-value vector $q \in \mathbb{R}^K$, let us sort it as $q_{(1)} \geq \cdots \geq q_{(K)}$. For $\pi \in \Delta^{K-1}$ and $M \geq 1$, define $S_i(q; \pi) := \sum_{j \in \text{Top}i(q)} \pi_j$, the policy mass on the top-$i$ indices of $q$. Then

$$J_{\text{ReMax}}^M(\pi, s, q) = q_{(1)} - \sum_{i=1}^{K-1} \left( q_{(i)} - q_{(i+1)} \right) \left( 1 - S_i(q; \pi) \right)^M. \tag{4}$$

Please refer to App. B.1 for the proof. The computational cost is $O(K \log K)$ to sort $q$ and does not depend on $M$. This closed-form expression allows us to calculate the inner expectation exactly, and it is differentiable with respect to $\pi$. Note that this relies on a (relatively small) discrete action space, where Q-values for all actions are available and we can sort them. For extension to continuous action spaces or vast discrete action spaces (e.g., language models), sample-based estimation is required.

**Base algorithm.** ReMax can be integrated into a broad family of RL algorithms that rely on Q-functions as critics. We highlight two classes: (1) **Actor–critic:** Actor–critic methods (Schulman et al., 2017; Haarnoja et al., 2018) are a natural fit since they already optimize policies with respect to critic signals. Thus, ReMax can be instantiated by replacing the policy-optimization module of the actor–critic algorithm by ReMax. (2) **Q-learning:** ReMax can also be combined with Q-learning variants (Mnih et al., 2013; Gallici et al., 2024; Vieillard et al., 2020), though it requires training a policy model in addition to the Q-function.

**Our instantiation.** Because ReMax is a novel policy-optimization objective, we adopt the simplest and most efficient instantiation: an actor–critic method with implicit Q-uncertainty modeling and the closed-form computation. Specifically, we use the on-policy actor-critic algorithm PPO (Schulman et al., 2017) as the base method, due to its strong performance in discrete-action settings. Note that we do not use the closed form exactly as in Eq. (4); rather, we derive a closed-form policy gradient, which shares the same spirit of leveraging sorting in Sec. 4. Other instantiations, such as explicit Q-distribution modeling, optimizing ReMax through Eq. (4), or integrating ReMax with other RL algorithms (e.g., Q-learning (Mnih et al., 2013)), are left for future work.

## 4 POLICY GRADIENT IN REMAX

To optimize the ReMax objective in Eq. (3) within on-policy actor–critic methods, we develop a practical policy gradient (PG) approach. As mentioned in the introduction to Q-function approximation in Sec. 3, only a single-trajectory return from $(s, a)$ is observable, motivating the design of a PG estimator based on such returns. In Sec. 4.1, we first show that a naïve PG derivation is not directly estimable from trajectory returns and then derive an estimation-friendly reformulation. We then provide a closed form of our proposed PG and generalize the number of draws to a positive real $m$ to provide fine-grained control over exploration in Sec. 4.2. Finally, we present an actor–critic instantiation, **Re**Max **PPO** (RePPO), based on Q-critic PPO (Schulman et al., 2017).

### 4.1 ESTIMATION-FRIENDLY POLICY GRADIENT FOR REMAX

Unbiased PG estimation is crucial for on-policy actor–critic. We therefore develop a PG for ReMax that can be estimated from trajectory returns, and first recall why standard RL admits the REINFORCE estimator based on trajectory returns.

**Policy Gradient in Standard RL.** Let $\pi_\theta$ be a parametrized policy, and define $J_{\mathrm{RL}}(\pi_\theta, s) := \mathbb{E}_{\tau \sim (\pi_\theta, P)} [\mathcal{R}(\tau) \mid s]$. The policy gradient theorem (Sutton & Barto, 2018) gives

$$\nabla_\theta J_{\mathrm{RL}}(\pi_\theta, s) = \mathbb{E}_{a \sim \pi_\theta} [\nabla_\theta \log \pi_\theta(a \mid s) Q^{\pi_\theta}(s, a)]. \tag{5}$$

Because the outer expectation is over a single action $\mathbb{E}_{a \sim \pi_\theta}[\cdot]$, replacing $Q^{\pi_\theta}(s, a)$ by the trajectory return $\mathcal{R}(s, a)$ yields an unbiased REINFORCE estimator: $\hat{g}_{\mathrm{RL}} := \nabla_\theta \log \pi_\theta(a \mid s) \mathcal{R}(s, a)$.

**Problem with the naïve PG for ReMax.** For ReMax with fixed Q-values $J_{\mathrm{ReMax}}^M(\pi_\theta, s, q)$, where $q \in \mathbb{R}^K$, applying the policy gradient theorem yields

$$\nabla_\theta J_{\mathrm{ReMax}}^M(\pi_\theta, s, q) = \mathbb{E}_{A_{1:M} \sim \pi_\theta} \left[ \max_{1 \le m \le M} q_{A_m} \sum_{m=1}^M \nabla_\theta \log \pi_\theta(A_m \mid s) \right]. \tag{6}$$

Unlike Eq. (5), the expectation is over $M$ actions, so an unbiased estimator would require observing returns for all $M$ sampled actions, which is infeasible in episodic RL. Moreover, $A_1, \ldots, A_M$ are coupled through the max operator, so a single-action expectation does not follow directly. We resolve this by introducing a baseline that decouples the max and enables a single-action expectation.

**Policy gradient via expected improvement.** For each $m$ in Eq. (6), we can insert a baseline $b_m$ that may depend on $(s, A_{-m})$ but not on $A_m$:

$$\nabla_\theta J_{\mathrm{ReMax}}^M(\pi_\theta, s, q) = \mathbb{E}_{A_{1:M} \sim \pi_\theta} \left[ \max_{1 \le m \le M} (q_{A_m} - b_m) \sum_{m=1}^M \nabla_\theta \log \pi_\theta(A_m \mid s) \right], \tag{7}$$

which preserves unbiasedness because $\mathbb{E}_{A_m} [\nabla_\theta \log \pi_\theta(A_m \mid s) b_m] = 0$. We choose $b_m$ as $W_{-m} := \max\{q_{A_1}, \ldots, q_{A_{m-1}}, q_{A_{m+1}}, \ldots, q_{A_M}\}$, then we have

$$\max_{1 \le j \le M} (q_{A_j} - W_{-m}) = (q_{A_m} - W_{-m})_+,$$

where $(x)_+ = \max(x, 0)$ for $x \in \mathbb{R}$. This turns the max into an action-specific term plus an "others" term, enabling the following single-action form:

---

**Algorithm 1** RePPO

---

1: **repeat**
2:     Collect trajectories under $\pi_\theta$ and compute returns $R_t^\lambda$.
3:     For each $(s_t, a_t)$: form $q \leftarrow Q_\phi(s_t, \cdot)$; compute $R_+(t) := \text{EI}_m(R_t^\lambda; \pi_\theta, q)$, $Q_+(s_t, a) = \text{EI}_m(Q_\phi(s_t, a); \pi_\theta, q)$, and advantage $A_+(t) = R_+(t) - b_+(s_t)$.
4:     Update actor by PPO objective (Eq. (12)) with $A_+(t)$; update critic $Q_\phi$ toward $R_t^\lambda$.
5: **until** convergence

---

**Proposition 2.** Let $W_{M-1} := \max\{q_{A_1}, \ldots, q_{A_{M-1}}\}$. Then

$$\nabla_\theta J_{\text{ReMax}}^M(\theta, s, q) = M \, \mathbb{E}_{a \sim \pi_\theta} \left[ \nabla_\theta \log \pi_\theta(a \mid s) \, \mathbb{E}_{A_{1:M-1} \sim \pi_\theta} \left[ (q_a - W_{M-1})_+ \right] \right]. \quad (8)$$

See App. B.2 for the proof. The blue term is the expected (negative) improvement of the maximum of the first $M-1$ draws over the action $a$. Since this quantity is central to our reformulated policy gradient, we refer to it as **Expected Improvement (EI)**, borrowing terminology from Bayesian optimization (Jones et al., 1998)[1]. For a reference $R \in \mathbb{R}$, policy $\pi$, and Q-values $q \in \mathbb{R}^K$, define $\text{EI}_M(R, \pi, q) := \mathbb{E}_{A_{1:M-1} \sim \pi} [(R - W_{M-1})_+]$, we have the EI-based PG.

**Definition 2** (EI-based PG). For fixed Q-values $q \in \mathbb{R}^K$, the EI-based PG is

$$\nabla_\theta J_{\text{ReMax}}^M(\theta, s, q) = M \, \mathbb{E}_{a \sim \pi_\theta} \left[ \nabla_\theta \log \pi_\theta(a \mid s) \, \text{EI}_M(q_a, \pi_\theta, q) \right]. \quad (9)$$

Taking the expectation over $q$ recovers the PG for ReMax in Eq. (3). Now, a single-trajectory return from $(s, a)$ yields an unbiased estimator: $\hat{g}_{\text{ReMax}} := \nabla_\theta \log \pi_\theta(a \mid s) \, \text{EI}_M(\mathcal{R}(s, a), \pi_\theta, q)$. (Walder & Karkhanis, 2025; Tang et al., 2025) also introduce a comparator baseline $W_{-m}$; however, they use it primarily for variance reduction. In contrast, we rewrite the REINFORCE-style PG with this baseline into the EI-based PG to enable estimation from trajectory returns. As you see in the following section, the EI also admits a closed-form computation as the ReMax objective does.

## 4.2 Efficient and Generalized Computation of EI

We provide a closed-form computation of the EI by leveraging the sorting of Q-values as in Prop. 1.

**Proposition 3** (Closed-form computation of EI). Let $q \in \mathbb{R}^K$ be Q-values at a state, $\pi \in \Delta^{K-1}$ a policy, $R \in \mathbb{R}$ a reference, and $M \in \mathbb{N}$. Define $v_i := (R - q_i)_+$ and sort $q$ as $q_{(1)} \geq \cdots \geq q_{(K)}$ with aligned masses $\pi_{(j)}$ and cumulatives $C_j = \sum_{u=1}^j \pi_{(u)}$. Then, we have

$$\text{EI}_M(R; \pi, q) = v_{(1)} + \sum_{j=1}^{K-1} \left( v_{(j+1)} - v_{(j)} \right) (1 - C_j)^{M-1}. \quad (10)$$

See App. B.3 for the proof. This also costs $\mathcal{O}(K \log K)$ time. Although ReMax is motivated by integer number of draws $M$, Eq. (10) remains well-defined for any real $m > 0$ by replacing $(1 - C_j)^{M-1}$ with $(1 - C_j)^{m-1}$. We therefore define the **generalized EI**, $\text{EI}_m(R; \pi, q)$, by substituting $m$ for $M$, enabling finer control of the exploration–exploitation trade-off. Please note that the closed-form of the ReMax with fixed Q-values (Prop. 1) is also valid for any real $m > 0$.

## 4.3 RePPO: Practical Policy Gradient for ReMax

We obtain a policy gradient that can be estimated directly from trajectory returns (Def. 2) and computed in closed form (Eq. (10)). Incorporating this into PPO (Schulman et al., 2017) leads to our new algorithm, **Re**Max **PPO** (RePPO). We begin by revisiting the PPO surrogate.

**PPO surrogate.** PPO collects trajectories $(s_0, a_0, \ldots, s_T)$ under the old policy $\pi_{\text{old}}$. For each state $s_t$, the $\lambda$-return $R_t^\lambda$ is computed, and the advantage is estimated with a value critic ($V_\phi : \mathcal{S} \to \mathbb{R}$)

---

[1]In Bayesian optimization, EI is the expected gain over the current best; here it is the improvement of an action over the best of $M-1$ other draws, a related but distinct notion.

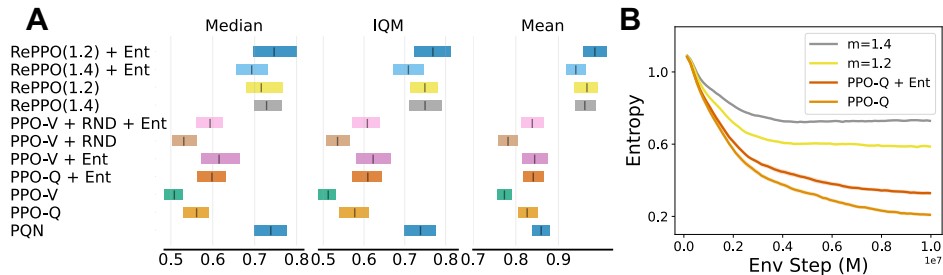

Figure 5: **A:** normalized scores aggregated with median, IQM, and mean across four games; boxes denote RLiable summaries over 10 seeds. **B:** policy entropy during Breakout training.

as $A(t) = R_t^\lambda - V_\phi(s_t)$. A straightforward policy gradient would differentiate $\pi_\theta(a_t \mid s_t) A(t)$. To improve sample efficiency, PPO reuses the same trajectory data for multiple updates. To stabilize training, it introduces the importance ratio $r_\theta(t) = \pi_\theta(a_t \mid s_t)/\pi_{\text{old}}(a_t \mid s_t)$ to add regularization toward the data-generating policy and a clip parameter $\varepsilon > 0$, yielding:

$$\mathcal{L}_{\text{PPO}}(\theta) := \mathbb{E}\left[\min\left(r_\theta(t)A(t),\ \text{clip}\left(r_\theta(t), 1-\varepsilon, 1+\varepsilon\right)A(t)\right)\right]. \tag{11}$$

**RePPO surrogate.** RePPO modifies PPO by two aspects: (1) replacing the value-critic with a Q-critic $Q_\phi$, which is necessary for ReMax; (2) introducing the EI-based advantage. At state $s_t$, for a computed $\lambda$-return $R_t^\lambda$, we construct an EI-based return as: $R_+(t) = \text{EI}_m(R_t^\lambda, \pi_\theta, Q_\phi(s_t, \cdot))$. The baseline can be obtained by taking the expectation of EI for all actions, approximated by $Q_+(s, a) = \text{EI}_m(Q_\phi(s, a), \pi_\theta, q)$ as $b_+(s) := \mathbb{E}_{a \sim \pi_\theta}[Q_+(s, a)]$. Then the EI-based advantage is given by $A_+(t) := R_+(t) - b_+(s_t)$. Finally, the surrogate objective for RePPO becomes

$$\mathcal{L}_{\text{RePPO}}(\theta) := \mathbb{E}\left[\min\left(r_\theta(t)A_+(t),\ \text{clip}\left(r_\theta(t), 1-\varepsilon, 1+\varepsilon\right)A_+(t)\right)\right]. \tag{12}$$

The algorithm is summarized in Alg. 1. Since RePPO differs from PPO solely through the use of a Q-critic and an EI-based advantage, its implementation is simple and the extra computation is minimal (see App. E.2). We provide the advantage computation code in App. D.

**Q-replacement for efficient exploration.** When we compute the EI for the return, $R_+(s, a) := \text{EI}_m(R(s, a); \pi_\theta, Q_\phi(s, \cdot))$, we expect $v_a = (R(s, a) - Q_\phi(s, a))_+$ to be zero in Eq. (10), since it measures the improvement of an action over itself. In practice, underspecified critics can yield $R(s, a) > Q_\phi(s, a)$, causing $R_+$ to be overestimated and the policy to overfit to $a$, which harms exploration. To mitigate this, when evaluating EI we *replace* the $a$-th element of $q = Q_\phi(s, \cdot)$ by $\mathcal{R}(s, a)$ ("Q-replacement").

## 5 EXPERIMENTS

We empirically evaluated RePPO on the MinAtar benchmark by answering two questions: **Q1:** Can RePPO, by optimizing ReMax, induce exploratory behavior and improve performance *without* exploration bonuses? **Q2:** Can a continuous retry parameter control exploration in a fine-grained manner and improve performance?

To confirm the emergence of exploration in RePPO, we used 3 benchmark environments: *MinAtar* (Young & Tian, 2019), *Atari* (Bellemare et al., 2013), and *Craftax* (Cobbe et al., 2020). MinAtar is a simplified version of Atari 2600 games providing Breakout, Asterix, Freeway, and Space Invaders. We used pgx implementation (Koyamada et al., 2023) for efficient vectorized simulation. Out of games in Atari, we used 10 games from Bellmare's hard-exploration problems (Bellemare et al., 2016) to verify the exploration benefits of RePPO. We reported the results in App. F. To further validate the scalability of RePPO, we used Craftax (Matthews et al., 2024), a vectorizable version of Crafter (Hafner, 2021), a open-ended RL environment.

### 5.1 MINATAR

**Baselines and hyperparameters.** We compared **RePPO** to: (i) **PPO-V** (standard PPO with a state-value critic (Schulman et al., 2017)); (ii) **PPO-Q** (PPO with a Q-critic); (iii) **PPO-V + RND** (PPO

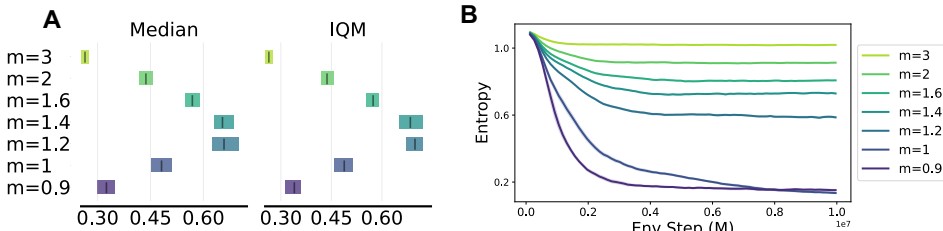

Figure 6: **Effect of retry parameter** $m$**: A:** median and IQM of normalized evaluation return across all games; **B:** policy entropy on Breakout. Larger $m$ slowed entropy decay and encouraged exploration; the best performance occurred around $m \in [1.2, 1.4]$.

with a Random Network Distillation (RND) (Burda et al., 2019)); (iv) **PQN** (Gallici et al., 2024), a strong Q-learning method for vectorized environments. PPO baselines followed the default `pgx` settings (based on PureJaxRL (Lu et al., 2022)). For PQN, we referred to the official implementations. To isolate the effect of entropy regularization, we reported results with and without entropy regularization for PPO and RePPO (+Ent indicates it was enabled). Main results used RePPO with $m = 1.2$ and $m = 1.4$. To compare with curiosity-based bonuses, we used Random Network Distillation (RND) (Burda et al., 2019) as a baseline. We also compare RND with and without entropy regularization. For speed comparisons of RePPO vs. PPO-V and PPO-Q, see App. E. For RePPO, we tuned $m$ and set $\lambda = 0.8$ for the $\lambda$-return, keeping all other hyperparameters identical to PPO-V. PQN's official setup used 128 parallel environments; we used 1024. To match the number of gradient updates, we adjusted the number of environments, minibatch size, and update epochs, and tuned $\lambda$ and the learning rate. Additional comparisons with the unmodified PQN hyperparameters (yielding ~5× more updates) are provided in App. E.2. Full hyperparameters are listed in App. E.1.

**Training and evaluation.** Agents trained for 10M environment steps with 10 seeds. At evaluation, we averaged 100 test episodes per seed. For PPO variants, during evaluation, the action was selected by argmax over policy logits; during training, we sampled from the policy. We reported *normalized scores* across games using median, IQM, and mean via RLiable (Agarwal et al., 2021), normalizing by the best score achieved among all methods. Per-game scores are in App. E.2.

**Main results.** Fig. 5 (A) aggregates performance via median, IQM, and mean across the four games. **RePPO** with $m \in [1.2, 1.4]$ achieved the strongest overall results among the PPO baselines (even with entropy or RND bonuses). While RePPO was comparable to PQN on median and IQM, it consistently outperformed PQN on mean, indicating its better tail performance. As shown in Fig. 5 (B), RePPO *without* entropy bonus maintained higher policy entropy than PPO *with* an entropy bonus, despite using no explicit bonus itself, which answers **Q1** in the affirmative. Furthermore, PPO-V + RND suffers from low performance when entropy is disabled, which indicates the importance of entropy regularization for RND and the effect of RePPO promoting exploration without entropy bonus.

**Effect of $m$: exploration–exploitation trade-off.** We tested $m \in \{0.9, 1.0, 1.2, 1.4, 1.6, 2, 3\}$ and reported (A) median and IQM of normalized scores and (B) policy entropy on Breakout (Fig. 6). The sweep showed that increasing $m$ systematically slowed entropy decay, with returns peaking near $m \approx 1.2$–$1.4$ and falling when $m$ was larger or smaller. Unlike integer-only $M$, the continuous parameter $m$ afforded precise tuning and yielded improved outcomes, answering **Q2** positively.

**Ablation on baseline and Q-replacement.** We further evaluated two key components of RePPO: the action-independent baseline and the Q-replacement strategy. With $m = 1.2$ without entropy bonus, we compared (i) *w/o base* (no baseline, Q-replacement enabled), (ii) *w/o rep* (baseline enabled, no Q-replacement), and (iii) *RePPO (full)* with both. Removing either component substantially degraded performance (Fig. 7), indicating that both are necessary to realize the full benefits of RePPO. Additional analysis of how Q-replacement preserves exploration is provided in App. E.2.

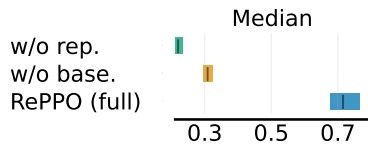

Figure 7: Median (all games).

## 5.2 CRAFTAX

Craftax (Matthews et al., 2024) is an open-ended RL environment built on top of Crafter (Hafner, 2021) and NetHack (Küttler et al., 2020), in which the agent must both plan over long horizons and continually adapt to newly revealed parts of the environment. We use the symbolic version of Craftax in our experiments.

**Setup.** As baselines, similar to the MinAtar experiment, we consider PPO-V, PPO-Q, and RND. For PPO-V, PPO-Q, and RND, we evaluate performance both with and without an entropy bonus (bonus coefficient is 0.01). RePPO is trained with $m = 1.2$ and $m = 1.4$, without using an entropy bonus. For all methods, we adopt the official implementation and hyperparameters released by the Craftax authors[2]. RePPO shares the same settings as the original implementation, except for its method-specific hyperparameters. Following the original paper, all agents are trained for 1 billion timesteps, and we report results at the end of training. Each configuration is trained with 5 random seeds, and after training, each seed is evaluated over 100 episodes. We report the mean and standard deviation of performance across seeds. As is standard practice, performance is reported as a percentage of the maximum achievable reward (226). The full set of hyperparameters is provided in App. G.

**Results.** The results are summarized in Table 1. RePPO (1.2), despite using neither an entropy bonus nor an intrinsic reward bonus, achieves performance competitive with PPO variants that rely on such bonuses, and clearly outperforms PPO without them. In App. G.1, we further report the evolution of policy entropy for each method, showing that—similarly to the MinAtar experiments—RePPO maintains higher entropy throughout training, even without explicit exploration bonuses. Notably, RND exhibits severe performance degradation when entropy regularization is disabled, highlighting the importance of entropy for RND and the capability of RePPO to encourage exploration without it. Overall, these results suggest that RePPO promotes exploration even in a larger-scale environment.

Table 1: Mean and std of the % of the max reward (226) over 5 seeds. Higher is better. RePPO (1.2) is **bolded** and is comparable to the PPO with entropy and RND bonuses (underlined).

| Method | % of max (std) |
|---|---|
| PPO-V | 9.31(1.00) |
| PPO-V + Ent | 11.66(0.33) |
| RND | 9.62(1.54) |
| RND + Ent | 11.68(0.30) |
| PPO-Q | 9.31(0.93) |
| PPO-Q + Ent | 11.85(0.26) |
| **RePPO (1.2)** | **11.87(0.16)** |
| RePPO (1.4) | 10.79(0.19) |

## 6 CONCLUSION

We introduced **ReMax**, a novel RL objective that encourages exploration by *directly* maximizing reward, without auxiliary bonus terms. We demonstrated its effectiveness as an exploration mechanism in bandit settings and extended the formulation to RL. To optimize ReMax efficiently, we derived a new policy–gradient expression and relaxed the retry count to a continuous parameter $m$, enabling fine-grained control over the exploration–exploitation trade-off. We instantiated these ideas in **RePPO**, a PPO-style deep actor–critic method with a Q-critic. On MinAtar, optimizing ReMax with RePPO induced exploratory behavior and improved performance without entropy bonuses.

**Limitations and Future Work.** Our study focuses on discrete-action problems, which allows efficient gradient computation (via sorting Q-values across actions) but limits immediate applicability to continuous action spaces. As discussed in Sec. 3, ReMax admits multiple integration paths in RL; exploring other bases, such as Q-learning (Mnih et al., 2013) and off-policy actor–critic (Haarnoja et al., 2018). Furthermore, explicitly modeling the epistemic uncertainty over Q-values by ensembles or Bayesian approaches is a promising direction. Finally, since ReMax exhibits empirically sublinear regret in bandits, establishing formal regret bounds would provide theoretical support for its exploratory benefits.

---

[2]https://github.com/MichaelTMatthews/Craftax_Baselines

**Reproducibility Statement.** We strive to make our results easy to reproduce. The ReMax objective, its closed-form components, and the policy-gradient estimator are specified in Secs. 3 and 4, with all assumptions and complete proofs in App. B. The exact advantage computation used in our implementation is provided in App. D. The MinAtar setup, environments, evaluation protocol, normalization, and all hyperparameters for every method, appears in Sec. 5 and App. E (including Tables and additional analyses). We report results over 10 random seeds and 100 evaluation episodes per seed and summarize performance with RLiable aggregates; per-game curves and ablations are in App. E.2. Complete details for the bandit experiments (problem setups, optimization procedure, and baselines) are in App. C. We will submit an anonymized archive of source code and configuration files as supplementary material, including scripts and instructions to reproduce all the results.

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

## A  EXTENDED RELATED WORK

This section provides a brief overview of exploration in RL.

**Optimism in the Face of Uncertainty (OFU).**    A prominent exploration strategy is OFU. Methods based on OFU mainly fall into two categories: confidence-based (Strehl & Littman, 2005; Jaksch et al., 2010; Dann & Brunskill, 2015) and bonus-based (Strehl et al., 2006; Strehl & Littman, 2008; Azar et al., 2017; Jin et al., 2018). While these approaches enjoy strong theoretical guarantees, they do not directly extend to deep RL, where explicit visitation counts are impractical to maintain. Bellemare et al. (2016) generalize visitation counts to enable OFU-style exploration in deep RL. There are several follow-ups that explores the count-based approach to deep RL, including Lobel et al. (2023); Ostrovski et al. (2017).

In contrast to OFU, ReMax does not employ an explicit exploration mechanism; instead, exploration emerges by maximizing a multiple-retry objective defined over Q-values.

**Intrinsic Motivation (IM).**    A complementary line of work that scales well with deep RL is intrinsic motivation (IM), which is broadly categorized into prediction-error-based, information-gain-based, and novelty-based methods.

Prediction-error-based methods learn a dynamics model and encourage visiting states (or state–action pairs) whose successor states are hard to predict (Stadie et al., 2015; Pathak et al., 2017); the idea dates back to Schmidhuber (1991b) and Thrun & Möller (1991). However, they can overemphasize inherently noisy states (Schmidhuber, 1991a). Information-gain-based methods instead seek states that reduce model uncertainty (Schmidhuber, 2010; Sun et al., 2011; Houthooft et al., 2016; Sukhija et al., 2024).

Novelty-based methods directly incentivize visiting "novel" states (or state–action pairs). Notions of novelty include pseudo-counts (Bellemare et al., 2016; Lobel et al., 2023; Taiga et al., 2021; Ostrovski et al., 2017), the estimated probability of a state appearing in a replay buffer (Fu et al., 2017), reachability-based metrics (Savinov et al., 2019), and intra-episode state diversity (Badia et al., 2020). Tang et al. (2017) discretize the state space with hashing to obtain counts.

These methods typically require estimating auxiliary density or dynamics models (e.g., transition models or visitation frequencies). By contrast, our method does not introduce any additional estimation targets.

Two notable exceptions that avoid explicit density estimation are RND (Burda et al., 2019) and E-values (Fox et al., 2018). Combining such novelty detectors to flag unexplored states and to intensify the use of ReMax for exploration is an interesting future direction.

**Entropy Maximization.**    Entropy-based exploration is widely used with policy-gradient and actor–critic methods (Williams & Peng, 1991; Mnih et al., 2016; Espeholt et al., 2018; Levine, 2018; Eysenbach & Levine, 2021). SAC is a popular example for continuous control (Haarnoja et al., 2018). While these approaches increase *action* entropy, increasing *state* entropy may align better with exploration goals (Pitis et al., 2020; Mutti et al., 2021), since action entropy alone promotes undirected exploration. Baram et al. (2021) propose maximizing transition entropy (the entropy of the next-state distribution given the current state) as a proxy for state entropy. In contrast, with ReMax, as shown in Sec. 2, directed (uncertainty-aware) exploration emerges from maximizing the retry objective.

**Posterior Sampling.**    Another family draws inspiration from Thompson sampling (Thompson, 1933; Honda & Takemura, 2014; Agrawal & Goyal, 2017). Bootstrapped DQN (Osband et al., 2016) samples a Q-function from an ensemble to emulate Thompson sampling, later enhanced with randomized priors and value functions (Osband et al., 2018; 2019; Ishfaq et al., 2021). Other works use Bayesian neural networks to model the Q-function posterior (Azizzadenesheli et al., 2018; Bayrooti et al., 2024; Osband et al., 2023) or adopt model-based posterior sampling (Sasso et al., 2023). Some works use Langevin Monte Carlo and efficient approximate sampling schemes to realize Thompson-style exploration in deep RL (Ishfaq et al., 2023; 2025). Integrating ReMax with such methods to *explicitly* model the Q-value distribution is a promising avenue.

**RL for LLM training.** Pass@K was introduced as an evaluation metric for LLM reasoning, measuring whether at least one of $K$ samples is correct (Chen et al., 2021). Beyond evaluation, recent work optimizes pass@K-like objectives directly. Walder et al. (Walder & Karkhanis, 2025) extend to continuous rewards via max@K, showing that maximizing the best of $K$ trials improves both pass@K and pass@1. Their formulation coincides with ours when per-action values are fixed. Tang et al. (Tang et al., 2025) analyze $K$-sample objectives (pass@K, majority vote), studying bias–variance and KL efficiency. Chen et al. (Chen et al., 2025) propose RLVR with pass@K as the training signal, using bootstrap grouping and analytic advantages to cut rollout cost. These works consistently show that retry-based training improves exploration and robustness in reasoning tasks. Despite sharing a similar idea, several key distinctions separate the scope of our work from theirs. **Uncertainty over rewards:** We model uncertainty over rewards or Q-values and adapt exploration to (epistemic) uncertainty, whereas LLM reasoning benchmarks typically assume a fixed reward. **Retry feasibility:** In episodic RL, multiple returns from the same state are generally infeasible; we emulate retries via a learned Q-function and derive a policy gradient estimable from single-trajectory data. **Novel PG with continuous retry:** Moreover, prior work controls retries with an integer $K$, whereas we introduce a continuous retry parameter, enabling fine-grained exploration–exploitation trade-offs. This extension was enabled by our novel policy gradient formulation and the closed-form expression as shown in Sec. 4. Thus, although max@K coincides with ReMax under fixed values, our contribution extends retry-based objectives to uncertainty-aware, episodic RL with continuous control of retries.

# B PROOFS

This section contains the proofs of the propositions and theorems in the main text.

## B.1 PROOF OF PROPOSITION 1

Here, using a sorting-based argument similar to Prop. 3, we can compute the ReMax objective exactly.

**Proposition 1.** Let $\mu = (\mu_1, \ldots, \mu_K)$ and write the order statistics $\mu_{(1)} \geq \cdots \geq \mu_{(K)}$. For $\pi \in \Delta_K$ and $M \geq 1$, define $S_i(\mu; \pi) := \sum_{j \in \text{Top}i(\mu)} \pi_j$, the policy mass on the top-$i$ indices of $\mu$. Then

$$J_{\text{ReMax}}^M(\pi, \mu) = \mu_{(1)} - \sum_{i=1}^{K-1} \left( \mu_{(i)} - \mu_{(i+1)} \right) \left( 1 - S_i(\mu; \pi) \right)^M. \tag{13}$$

*Proof.* Fix $\mu$. For $i \in \{1, \ldots, K-1\}$, define the event

$$F_i := \left\{ \text{none of } A^{(1)}, \ldots, A^{(M)} \text{ falls in } \text{Top}_i(\mu) \right\}. \tag{14}$$

Equivalently, $F_i$ is the event that all $M$ draws miss the top-$i$ arms under $\mu$. Define the (random) rank of the best arm that is hit at least once by

$$r^\star := \min r \in 1, \ldots, K : \exists m \in 1, \ldots, M \text{ with } A^{(m)} \in \text{Top}_r(\mu), \tag{15}$$

with the convention $r^\star = K$ if no draw hits any of the top-$(K-1)$ arms. By construction, $F_i$ occurs iff $i < r^\star$, and does not occur iff $i \geq r^\star$.

Consider the following pathwise (deterministic) identity:

$$\max_{1 \leq m \leq M} \mu_{A^{(m)}} = \mu_{(1)} - \sum_{i=1}^{K-1} \left( \mu_{(i)} - \mu_{(i+1)} \right) \mathbf{1}F_i. \tag{16}$$

To verify (16), note that on the event $r^\star = r$ the right-hand side equals

$$\mu_{(1)} - \sum_{i=1}^{r-1} \left( \mu_{(i)} - \mu_{(i+1)} \right) = \mu_{(r)} = \max_{1 \leq m \leq M} \mu_{A^{(m)}}. \tag{17}$$

Thus (16) holds almost surely (i.e., for every outcome of the $M$ draws).

Taking conditional expectations of (16) given $\mu$ yields

$$\mathbb{E}_{A^{(1:M)}\sim\pi}\left[\max_{1\leq m\leq M}\mu_{A^{(m)}}\Big|\mu\right] = \mu_{(1)} - \sum_{i=1}^{K-1}\big(\mu_{(i)}-\mu_{(i+1)}\big)\mathbb{P}_{A^{(1:M)}\sim\pi}[F_i \mid \mu]. \quad (18)$$

By independence and identical distribution of the draws, a single draw hits $\mathrm{Top}_i(\mu)$ with probability $S_i(\mu;\pi)$, hence

$$\mathbb{P}_{A^{(1:M)}\sim\pi}[F_i \mid \mu] = \big(1 - S_i(\mu;\pi)\big)^M. \quad (19)$$

Plugging this into the previous display completes the proof. $\qquad\square$

### B.2 Proof of Proposition 2.

**Proposition 2.** Let $W_{M-1} := \max\{q_{A_1},\ldots,q_{A_{M-1}}\}$, we have:

$$\nabla_\theta J^M_{\mathrm{ReMax}}(\theta,s,q) = M\mathbb{E}_{a\sim\pi_\theta}\left[\nabla_\theta\log\pi_\theta(a|s)\mathbb{E}_{A_{1:M-1}}\left[(q_a-W_{M-1})_+\right]\right]. \quad (20)$$

*Proof.* We start by considering the PG with a per-term baseline.

$$\nabla_\theta J_M(\pi_\theta,s,q) = \mathbb{E}_{A_{1:M}}\left[\sum_{m=1}^M\nabla_\theta\log\pi_\theta(A_m)\left(\max_{1\leq j\leq M}q_{A_j} - b_m\right)\right]. \quad (21)$$

Choose $b_m$ as

$$b_m := W_{-m} := \max\{q_{A_1},\ldots,q_{A_{m-1}},q_{A_{m+1}},\ldots,q_{A_M}\}, \quad (22)$$

which preserves unbiasedness since

$$\mathbb{E}_{A_m\sim\pi_\theta}\left[\nabla_\theta\log\pi_\theta(A_m)\,b_m\right] = b_m\,\nabla_\theta\sum_{i=1}^K\pi_\theta(i) = 0, \quad (23)$$

With this baseline,

$$\max_{1\leq j\leq M}q_{A_j} - W_{-m} = \big(q_{A_m} - W_{-m}\big)_+. \quad (24)$$

Hence

$$\nabla_\theta J_M(\pi_\theta,s,q) = \mathbb{E}_{A_{1:M}}\left[\sum_{m=1}^M\nabla_\theta\log\pi_\theta(A_m)(q_{A_m} - W_{-m})_+\right]. \quad (25)$$

**Condition on the first $M-1$ samples.** By symmetry of i.i.d. draws,

$$\nabla_\theta J_M(\pi_\theta,s,q) = M\,\mathbb{E}_{q,A_{1:M-1}}\left[\mathbb{E}_{A_M\sim\pi_\theta}\left[\nabla_\theta\log\pi_\theta(A_M)\,(q_{A_M} - W_{M-1})_+\right]\right], \quad (26)$$

where

$$W_{M-1} := \max\{q_{A_1},\ldots,q_{A_{M-1}}\}. \quad (27)$$

For fixed $(q,A_{1:M-1})$, $W_{M-1}$ is a constant and $A_M\sim\pi_\theta$, so

$$\mathbb{E}_{A_M\sim\pi_\theta}\left[\nabla_\theta\log\pi_\theta(A_M)\,(q_{A_M} - W_{M-1})_+\right] = \sum_{i=1}^K\pi_\theta(i)\,\nabla_\theta\log\pi_\theta(i)\,(q_i - W_{M-1})_+. \quad (28)$$

**Separate the action expectation.** Then, we have

$$\nabla_\theta J_M(\pi_\theta,s,q) = M\,\mathbb{E}_{i\sim\pi_\theta}\left[\mathbb{E}_{A_{1:M-1}\sim\pi_\theta}\left[\nabla_\theta\log\pi_\theta(i)\,(q_i - W_{M-1})_+\right]\right], \quad (29)$$

which completes the proof. $\qquad\square$

### B.3 PROOF OF PROPOSITION 3.

**Proposition 3.** Let $q \in \mathbb{R}^K$ be Q-values at a state, $\pi \in \Delta^{K-1}$ a policy, $R \in \mathbb{R}$ a reference, and $M \in \mathbb{N}$. Define $v_i := (R - q_i)_+$ and sort $q$ as $q_{(1)} \geq \cdots \geq q_{(K)}$ with aligned masses $\pi_{(j)}$ and cumulatives $C_j = \sum_{u=1}^{j} \pi_{(u)}$. Then

$$\text{EI}_M(R; \pi, q) = v_{(1)} + \sum_{j=1}^{K-1} \left( v_{(j+1)} - v_{(j)} \right) (1 - C_j)^{M-1}. \tag{30}$$

*Proof.* Remember that the expected improvement is defined as

$$\text{EI}_M(R; \pi, q) := \mathbb{E}_{A_1,\ldots,A_M \sim \pi} \left[ \left( R - \max_{1 \leq m \leq M} q_{A_m} \right)_+ \right], \tag{31}$$

where $(x)_+ := \max(x, 0)$. Then, for the transformed values $v_i := (R - q_i)_+$, this can be written as

$$\text{EI}_M(R; \pi, q) = \mathbb{E}_{A_1,\ldots,A_M \sim \pi} \left[ \min_{1 \leq m \leq M} v_{A_m} \right]. \tag{32}$$

**Telescoping form.** Sort the transformed values in ascending order,

$$v_{(1)} \leq v_{(2)} \leq \cdots \leq v_{(K)}, \tag{33}$$

and let $\pi_{(j)}$ be the policy mass corresponding to $v_{(j)}$. Define the cumulative masses

$$C_j := \sum_{m=1}^{j} \pi_{(m)}, \qquad j = 1, \ldots, K, \qquad C_0 := 0. \tag{34}$$

Then the expected improvement can be written as:

$$\text{EI}_M(R; \pi) = v_{(1)} + \sum_{j=1}^{K-1} \left( v_{(j+1)} - v_{(j)} \right) (1 - C_j)^{M-1}, \tag{35}$$

which completes the proof. □

## C   DETAILS OF THE BANDIT EXPERIMENTS

This appendix provides details of the bandit experiments.

### C.1   BINARY BANDIT

We can compute the ReMax objective for the binary bandit analytically as follows:

$$J_{\text{ReMax}}^M(p) := 0.75 \cdot (1 - (1 - p)^M) + 0.25 \cdot (1 - p^M), \tag{36}$$

where $p$ is the probability of pulling arm 1.

### C.2   BERNOULLI BANDIT

We describe (i) the Bernoulli-bandit setup, (ii) the experimental design for Fig. 2, and (iii) the computations for each method.

**(i) Bernoulli-bandit setup.** We consider two arms with rewards $R_i = \alpha_i X_i$, where $X_i \sim \text{Bernoulli}(p_i)$ and $\mu_i = \mathbb{E}[R_i] = \alpha_i p_i$. For a fixed realization $r = (r_0, r_1)$ and a policy $\pi \in [0, 1]$ denoting $\Pr(a{=}0) = \pi$ (thus $\Pr(a{=}1) = 1 - \pi$), the ReMax objective with $M{=}2$ i.i.d. draws is

$$J^2(\pi \mid r) = \pi^2 r_0 + 2\pi(1 - \pi) \max\{r_0, r_1\} + (1 - \pi)^2 r_1.$$

Taking expectation over Bernoulli outcomes $e_1 = (\alpha_0, 0)$, $e_2 = (0, \alpha_1)$, $e_3 = (\alpha_0, \alpha_1)$, $e_4 = (0, 0)$ with probabilities $p_{e_1} = p_0(1 - p_1)$, $p_{e_2} = (1 - p_0)p_1$, $p_{e_3} = p_0 p_1$, $p_{e_4} = (1 - p_0)(1 - p_1)$ gives

$$\mathbb{E}\left[ J^2(\pi \mid R) \right] = \sum_{k=1}^{4} p_{e_k} J^2(\pi \mid e_k).$$

**(ii) Experimental design.** We fix $p_0 = 1$ and $\alpha_0 = 2$, and sweep arm-1 scale $\alpha_1 \in [1, 10]$ while adjusting $p_1$ to keep the mean constant: $\alpha_1 p_1 = 1$. For each $\alpha_1$, we evaluate $\pi^\star(a=1) = 1 - \pi^\star$ under $M = 2$ and plot it together with the softmax baseline in Fig. 2.

**(iii) Method computations.** *ReMax (M=2).* Compute $\mathbb{E}\left[J^2(\pi \mid R)\right]$ via the above event decomposition and maximize over $\pi \in [0, 1]$ (we use a dense numerical search; a closed form exists because the objective is quadratic in $\pi$).
*Softmax (entropy-regularized).* For temperature $\beta > 0$ (we use $\beta=1$),

$$\pi_{\text{soft}}(a=1) = \frac{\exp(\mu_1/\beta)}{\exp(\mu_0/\beta) + \exp(\mu_1/\beta)}, \quad \mu_0 = \alpha_0 p_0, \ \mu_1 = \alpha_1 p_1 \ (= 1 \text{ in our sweep}).$$

### C.3 BANDIT WITH POSTERIOR

---
**Algorithm 2** ReMax with Posterior

---
1: Initialize prior $\Pi_{0,i}$ for each arm.
2: **for** $t = 1, 2, \ldots, T$ **do**
3:     Compute $\pi_t = \arg\max_\pi \mathbb{E}_{\mu_t \sim \Pi_t}\left[J^M_{\text{ReMax}}(\pi; \mu_t)\right]$ by Alg. 3.
4:     Play $a_t \sim \pi_t$, observe $r_t \in \{0, 1\}$.
5:     Update posterior of arm $a_t$, $\Pi_{t+1,a_t}$, by the posterior update rule.
6: **end for**

---

---
**Algorithm 3** ReMax optimization

---
**Require:** Posterior $\Pi_t$, batch size $B$, draws $M$, epochs $S$ and policy $\theta_{t-1}$
1: Initialize policy $\pi_\theta$ with $\theta_{t-1}$
2: **for** $s = 1, 2, \ldots, S$ **do**
3:     Sample $\mu^{b,t} \sim \Pi_t$ for $b = 1, 2, \ldots, B$.
4:     Compute $J^M_{\text{ReMax}}(\pi_\theta; \mu^{b,t})$ for $b = 1, 2, \ldots, B$.
5:     Update $\pi_\theta$ by gradient descent.
6: **end for**
7: **return** $\pi_t = \pi_\theta$

---

To confirm empirically sublinear regret, we conduct a posterior-driven bandit experiment in two settings: Beta–Bernoulli and Gaussian–Gaussian. We consider a ground-truth prior $\Pi^*$ over the arm means $\{\mu_i\}_{i=1}^K$. Initially, each $\mu_i \sim \Pi^*$ and the learner's prior is $\Pi_0 = \Pi^*$. At each round $t$, we select $a_t$, observe $r_t$ drawn with mean $\mu_{a_t}$, and update the posterior $\Pi_{t+1,a_t}$.

**Beta–Bernoulli.** The prior $\Pi^*$ is $\text{Beta}(\alpha_0, \beta_0)$ with $\alpha_0 = \beta_0 = 1$ for all arms. Rewards are Bernoulli with mean $\mu_i$. The posterior update is

$$\Pi_{t+1,i} = \text{Beta}(\alpha_t + r_t, \ \beta_t + 1 - r_t).$$

**Gaussian–Gaussian.** The prior $\Pi^*$ is $\mathcal{N}(\mu_0, \sigma_0^2)$ with $\mu_0 = 0$ and $\sigma_0^2 = 1$ for all arms. Rewards are $\mathcal{N}(\mu_i, \sigma_R^2)$ with $\sigma_R^2 = 1$. The posterior update is

$$\Pi_{t+1,i} = \mathcal{N}(\mu_{i,t+1}, \sigma_{i,t+1}^2), \qquad \sigma_{i,t+1}^2 = \left(\tfrac{1}{\sigma_{i,t}^2} + \tfrac{1}{\sigma_R^2}\right)^{-1}, \quad \mu_{i,t+1} = \sigma_{i,t+1}^2 \left(\tfrac{\mu_{i,t}}{\sigma_{i,t}^2} + \tfrac{r_t}{\sigma_R^2}\right).$$

**ReMax optimization.** Arm selection by ReMax is shown in Alg. 2. At each $t$, we optimize the ReMax objective via Alg. 3, the exact computation of the objective from Prop. 1 with batch size $B = 16$ and epochs $S = 50$. For ease of optimization, at each round $t$ we initialize the policy $\theta_t$ with the parameters from the previous round $t - 1$. To avoid optimizer state carrying over across rounds, we reinitialize the optimizer at each round.

**Baselines.** We compare against Thompson sampling (Thompson, 1933; Honda & Takemura, 2014; Agrawal & Goyal, 2017) and UCB (Auer et al., 2002), both with sublinear-regret guarantees. Thompson sampling: sample $\mu_a$ from $\Pi_{t,a}$ and select $a = \arg\max_a \mu_a$. UCB: after initializing by pulling all arms, select the arm with highest empirical mean plus a bonus $c\sqrt{\log(t)/(2N_a)}$, where $N_a$ is the number of pulls; we use $c = 1.0$ for both settings. To compare with the entropy-regularized exploration, we prepared a Softmax baseline, where we took the softmax of the posterior means of each arm and selected the arm following the softmax distribution. We fine-tune the temperature parameter in $(0.01, 0.1, 1.0)$ and used $0.1$ for the experiments.

## D   RePPO

Below is the code for the advantage computation in RePPO.

Listing 1: Advantage Computation for RePPO

```
def expected_improvement_min(
    R: jnp.ndarray,  # (B, N_ref)
    q: jnp.ndarray,  # (B, K)
    pi: jnp.ndarray, # (B, K)
    M: float,
) -> jnp.ndarray:
    """EI_M(R;pi) = E[min_{1..M} (R - q_A)_+],  A~pi i.i.d.
       Returns: (B, N_ref)
    """
    idx = jnp.argsort(-q, axis=-1)
    q_sorted = jnp.take_along_axis(q, idx, axis=-1)
    pi_sorted = jnp.take_along_axis(pi, idx, axis=-1)

    C = jnp.cumsum(pi_sorted, axis=-1)  # (B, K)

    v = jnp.maximum(R[..., None] - q_sorted[:, None, :], 0.0)  # (B,
        N_ref, K)
    v_first = v[..., 0]
    dv = v[..., 1:] - v[..., :-1]
    eps = 1e-8
    w = jnp.power(jnp.clip(1.0 - C[..., :-1], eps, 1.0), M)  # (B, K-1)
    EI = v_first + jnp.sum(dv * w[:, None, :], axis=-1)      # (B, N_ref)
    return EI

def reppo_advantage(
    R: jnp.ndarray,        # (B,) or (B,1)
    q: jnp.ndarray,        # (B, K)
    pi: jnp.ndarray,       # (B, K)
    action: jnp.ndarray,   # (B,)
    M: float,  # This is actually M-1 in our paper
) -> jnp.ndarray:
    """Compute RePPO advantage with Q-replacement by return.
       Returns: (B,)
    """
    if R.ndim == 1:
        R = R[:, None]  # (B, 1)

    # Q-replacement: set q[action] <- R
    q_ref = q.at[jnp.arange(q.shape[0]), action].set(R[:, 0])

    # Improvement terms
    R_plus = expected_improvement_min(R, q_ref, pi, M)[..., 0]  # (B,)
    q_plus = expected_improvement_min(q, q_ref, pi, M)          # (B, K)
    baseline = jnp.sum(pi * jax.lax.stop_gradient(q_plus), axis=-1)  # (B
        ,)
    return R_plus - baseline
```

# E  MINATAR EXPERIMENT

In this appendix, we provide additional details on the MinAtar experiment.

## E.1  EXPERIMENTAL SETUP

**Network architecture.**  We use the same network architecture as the public implementations of PPO and PQN (official implementation). **PPO (Actor–Critic).** A shared CNN (Conv $2\times2$ + ReLU + avg-pool) and MLP produce a latent state, which branches into (i) an actor head with two hidden layers (ReLU/Tanh) outputting action logits, and (ii) a critic head with two hidden layers outputting per-action Q-values (i.e., a Q-critic instead of a scalar $V$). **PQN (Q-Network).** Inputs are scaled (optional BatchNorm), then passed to a CNN feature extractor (Conv $3\times3$ with LayerNorm/BatchNorm + ReLU) and an MLP; a final linear layer outputs Q-values for all actions. This is a single-head, value-based model tailored to pure Q-learning.

**Code references.**  We list the code references used in our experiments.

- MinAtar: `pgx` implementation (Koyamada et al., 2023)[3]

- PPO: `pgx` implementation (Koyamada et al., 2023)[4], which is also based on `purejaxrl` (Lu et al., 2022)[5]

- PQN: official implementation (Gallici et al., 2024)[6]

**Hyperparameters.**  For PQN with 1024 parallel environments, we tuned the learning rate (0.0005, 0.001) and GAE $\lambda$ (0.65, 0.8, 0.95) for each environment. For RND, we tuned the learning rate (0.001, 0.0003, 0.0001) of the RND network and bonus coefficient (0.5, 1.0, 1.5) for each environment. We report the hyperparameters for RePPO, PPO-V, PPO-Q, and PQN in Tables 2, 3, 4, and 5, respectively.

Table 2: Hyperparameters for RePPO.

| Name (symbol) | Value |
|---|---|
| total timesteps | $1.0 \times 10^7$ |
| learning rate | $3.0 \times 10^{-4}$ |
| rollout length | 128 |
| parallel envs | 1024 |
| update epochs | 3 |
| minibatch size | 1024 |
| discount $\gamma$ | 0.99 |
| GAE $\lambda$ | 0.8 |
| clip $\varepsilon$ | 0.2 |
| entropy coefficient | (0.0, 0.01) |
| value loss coefficient | 0.5 |
| max grad norm | 0.5 |
| optimizer | Adam (with global-norm clip) |
| RePPO draws $m$ | (1.2, 1.4) |

---

[3]https://github.com/sotetsuk/pgx

[4]https://github.com/sotetsuk/pgx

[5]https://github.com/luchris429/purejaxrl

[6]https://github.com/mttga/purejaxql

Table 3: Hyperparameters for PPO-V and PPO-Q.

| Name (symbol) | Value |
|---|---|
| total timesteps | $1.0 \times 10^7$ |
| learning rate | $3.0 \times 10^{-4}$ |
| rollout length | 128 |
| parallel envs | 1024 |
| update epochs | 3 |
| minibatch size | 1024 |
| discount $\gamma$ | 0.99 |
| GAE $\lambda$ | 0.95 |
| clip $\varepsilon$ | 0.2 |
| entropy coefficient | (0.0, 0.01) |
| value loss coefficient | 0.5 |
| max grad norm | 0.5 |
| optimizer | Adam (with global-norm clip) |
| **RND related** | |
| learning rate | 0.001 (Asterix), 0.0003 (Breakout, Freeway), 0.0001 (Space Invaders) |
| bonus coefficient | 1.5 (Freeway), 1.0 (others) |

Table 4: Original hyperparameters for PQN.

| Name | Value |
|---|---|
| total timesteps | $1.0 \times 10^7$ |
| parallel envs | 128 |
| rollout length | 32 |
| minibatch size | 128 |
| update epochs | 2 |
| learning rate | $5.0 \times 10^{-4}$ |
| LR linear decay | `True` |
| max grad norm | 10.0 |
| discount $\gamma$ | 0.99 |
| GAE $\lambda$ | 0.65 |
| $\varepsilon$-start / finish / decay ratio | 1.0 / 0.05 / 0.1 |
| normalization type | `layer_norm` |
| optimizer | RAdam (with global-norm clip) |

Table 5: Tuned hyperparameters for PQN (1024 parallel environments)

| Name | Value |
|---|---|
| total timesteps | $1.0 \times 10^7$ |
| parallel envs | 1024 |
| rollout length | 128 |
| minibatch size | 1024 |
| update epochs | 3 |
| learning rate | $1.0 \times 10^{-3}$ |
| LR linear decay | `True` |
| max grad norm | 10.0 |
| discount $\gamma$ | 0.99 |
| GAE $\lambda$ | 0.8 |
| $\varepsilon$-start / finish / decay ratio | 1.0 / 0.05 / 0.1 |
| normalization type | `layer_norm` |
| optimizer | RAdam (with global-norm clip) |

**Training and evaluation.** Agents are trained for 10M environment steps with 10 random seeds. During evaluation, we average 100 test episodes per seed. For PPO variants, evaluation uses the

argmax of the policy's logits, whereas training samples actions from the policy. We report *normalized scores* aggregated across games using median, interquartile mean (IQM), and mean, following the RLiable framework (Agarwal et al., 2021). Scores are normalized by the maximum score achieved across all methods. Per-game scores are reported in App. E.2. The maximum scores used for normalization are: Breakout 251.15, Asterix 64.95, Freeway 67.05, Space Invaders 880.91.

## E.2 ADDITIONAL RESULTS

We present supplementary analyses to complement the main results. Unless stated otherwise, curves show the mean and standard error over 10 seeds.

**Per-game results.** Fig. 8 reports learning curves for each MinAtar game. RePPO clearly outperforms PPO on *Breakout* and *Asterix*, and converges faster than the baselines on *Freeway*, consistent with the main findings. On *Asterix*, PQN outperforms all other methods.

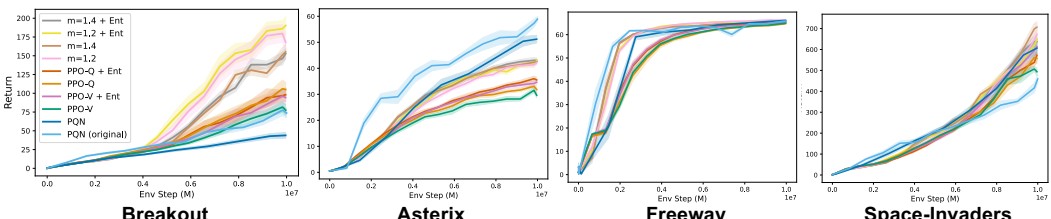

Figure 8: Per-game learning curves. Mean ± s.e. over 10 seeds.

**Comparison to the original PQN.** In the main text, we compared RePPO to PQN tuned for our setup. Fig. 9 adds results for the original PQN. Overall, the original PQN is comparable to the one that we tuned for our setup. PQN is comparable to RePPO on Median and IQM, but underperforms on Mean, the same trend we observed in the main text. This confirms that the difference of the setup does not effect the performance of PQN that much, confirming the validity of our analysis.

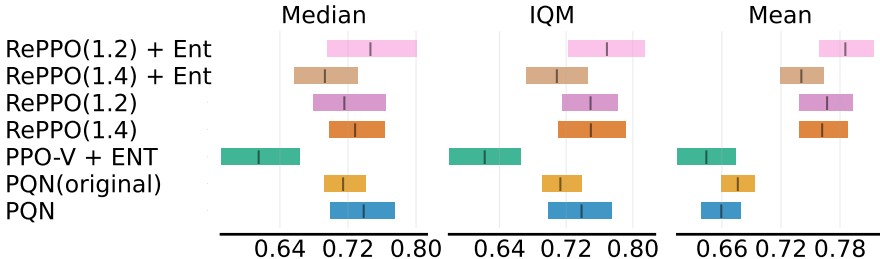

Figure 9: Aggregate metrics (Median, IQM, Mean) across all games.

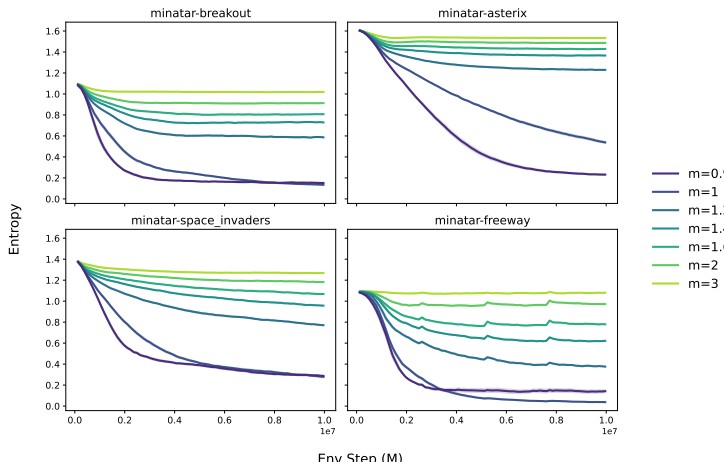

Figure 10: Policy entropy for all environments over $m \in \{0.9, 1.0, 1.2, 1.4, 1.6, 2, 3\}$.

**Entropy across environments.** Fig. 10 shows policy entropy for all environments and $m \in \{0.9, 1.0, 1.2, 1.4, 1.6, 2, 3\}$. As expected, entropy aligns with the retry parameter $m$: small $m$ leads to rapid entropy decay, while larger $m$ sustains higher entropy. Entropy can also be tuned more finely by varying $m$ between these values.

**Details on the analysis of Q-replacement.** We hypothesize that Q-replacement mitigates overestimation of Expected Improvement (EI), which would otherwise reduce exploration. To test this, we compare entropy with and without Q-replacement on *Breakout* (Fig. 11). The no-replacement variant exhibits consistently lower entropy, confirming that Q-replacement preserves exploration and contributes to performance.

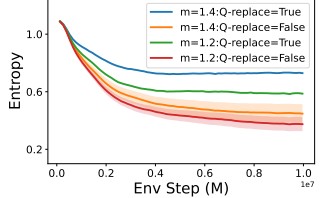

Figure 11: Entropy.

**Speed comparison.** We compare the training speed of RePPO, PPO-V, and PPO-Q in MinAtar. Fig. 12 reports the average wall-clock time on *Breakout* for 10M timesteps (no evaluation), averaged over 5 seeds. Hyperparameters match Sec. 5. Since our implementation uses the JAX framework, we exclude JIT warmup time for all methods. As expected, RePPO is slower than PPO-V and PPO-Q because it computes EI for the advantage. However, the additional cost is comparable to the gap between PPO-V and PPO-Q, that is, to the overhead incurred by replacing a $V$-critic with a $Q$-critic. This suggests that the additional cost of RePPO, such as sorting $Q$-values and computing EI, is negligible compared to the improvement in performance.

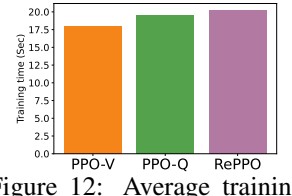

Figure 12: Average training time (Breakout).

**The standard deviation of the EI-based advantage.** Fig. 13 shows the standard deviation of the EI-based advantage for Breakout during training. Error bars represent the standard error across 5 random seeds. The standard deviations for exploratory retry parameters (e.g., $m = 1.2$ and $m = 1.4$) are lower than those for non-exploratory parameters (e.g., $m = 0.9$).

## F  ATARI EXPERIMENT

We use the 10 Atari environments identified as dense-reward, hard-exploration problems by Bellemare et al. (Bellemare et al., 2016), namely: Alien, Amidar, BattleZone, Frostbite, Hero, Ms. Pacman, Q*bert, Surround, Wizard of Wor, and Zaxxon. Our implementation is based on the CleanRL codebase (Huang et al., 2022) and employs EnvPool (Weng et al., 2022) for parallel environment execution.

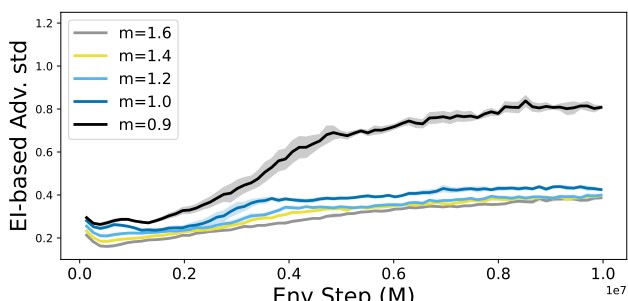

Figure 13: The standard deviation of the EI-based advantage.

## F.1 EXPERIMENTAL SETUP

Table 6: Hyperparameters for RePPO.

| Name (symbol) | Value |
| --- | --- |
| total timesteps | $1.0 \times 10^7$ |
| learning rate | $2.5 \times 10^{-4}$ |
| rollout length | 128 |
| parallel envs | 128 |
| update epochs | 4 |
| minibatch size | 128 |
| discount $\gamma$ | 0.99 |
| GAE $\lambda$ | 0.95 |
| clip $\varepsilon$ | 0.1 |
| entropy coefficient | (0.0, 0.01) |
| value loss coefficient | 0.5 |
| max grad norm | 0.5 |
| optimizer | Adam (with global-norm clip) |
| RePPO draws $m$ | (0.8, 0.9, 1.0, 1.2, 1.4) |

Table 7: Hyperparameters for PPO-V and PPO-Q.

| Name (symbol) | Value |
| --- | --- |
| total timesteps | $1.0 \times 10^7$ |
| learning rate | $2.5 \times 10^{-4}$ |
| rollout length | 128 |
| parallel envs | 128 |
| update epochs | 4 |
| minibatch size | 128 |
| discount $\gamma$ | 0.99 |
| GAE $\lambda$ | 0.95 |
| clip $\varepsilon$ | 0.1 |
| entropy coefficient | (0.0, 0.01) |
| value loss coefficient | 0.5 |
| max grad norm | 0.5 |
| optimizer | Adam (with global-norm clip) |

We follow the hyperparameters listed in Tables 6 and 7, which are adapted from CleanRL. As baselines, we use PPO-V and PPO-Q, each evaluated with and without entropy regularization. RePPO is trained with retry parameters $m = 0.8, 0.9, 1.0, 1.2, 1.4$, and we report results for all configurations. Training is conducted for $1 \times 10^7$ environment steps.

Evaluation is carried out in parallel with training using eight environments, and we report normalized scores at the final training step. All results are averaged over five random seeds and aggregated

across the 10 environments using the RLiable framework (Agarwal et al., 2021) to compute the median, interquartile mean (IQM), and mean performance and it is shown in Fig. 14. We also plotted the raw return curves for all games in Fig. 15.

## F.2 RESULTS

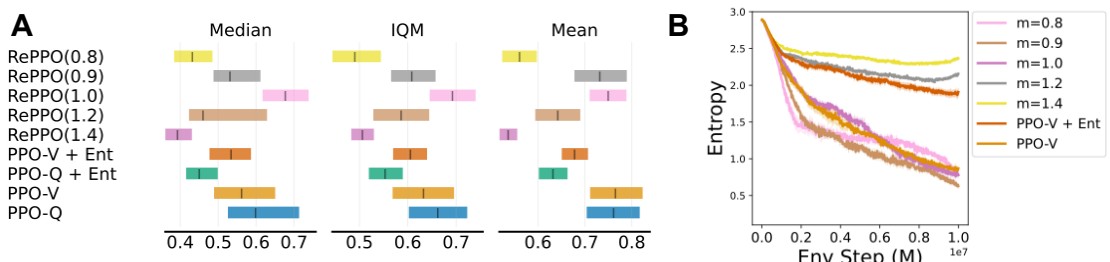

Figure 14: **A:** normalized scores aggregated with median, IQM, and mean across 10 games; boxes denote RLiable summaries over 5 seeds. For RePPO, we see the performance peak around $m = 0.9$ to 1.0, and PPO without entropy performs the better than that with entropy. This indicates that those environments indeed require less exploration. **B:** policy entropy during Alien training.

Fig. 14 presents the experimental results. Panel A shows that, across all environments, methods with weaker exploration, such as PPO-V, PPO-Q, and RePPO with $m = 0.9$ or 1.0, achieve the highest performance. In contrast, RePPO with larger retry parameters ($m = 1.2$ and 1.4) and entropy-regularized PPO variants exhibit lower performance. This results in a performance peak around $m = 0.9$ to 1.0, a trend that also appeared in the MinAtar experiments. These findings suggest that, in practice, Bellemare's suite of hard-exploration environments may require relatively little exploration. Panel B shows the evolution of policy entropy during training on Alien and in Fig. 16, we show the policy entropy on all games.

For $m = 1.2$ and 1.4, the policy maintains higher entropy, indicating that RePPO indeed encourages more exploratory behavior even in complex, pixel-based environments such as Atari. Taken together, these results demonstrate that RePPO promotes exploration in Atari, but also remains flexible: when little exploration is needed, choosing a smaller retry parameter (e.g., $m = 0.9$ or 1.0) yields strong performance. In contrast, larger values encourage exploration when desired.

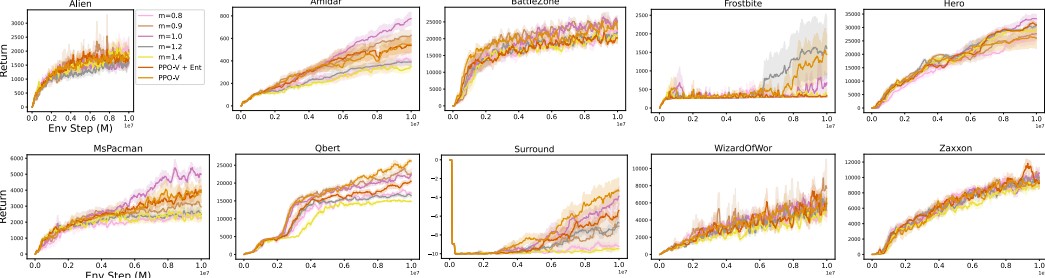

Figure 15: Plot of the return curves for all games. Mean $\pm$ s.e. over 5 seeds. Overall, PPO without entropy and RePPO with $m = 0.9$ and 1.0 performs better than that with entropy and RePPO with $m = 1.2$ and 1.4. This indicates that those environments indeed require less exploration.

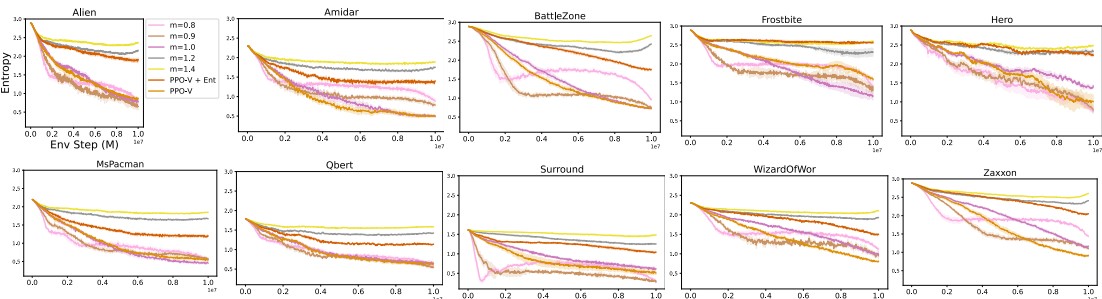

Figure 16: Policy entropy during training on all games. Mean $\pm$ s.e. over 5 seeds. RePPO with $m = 1.2$ and $1.4$ maintains higher entropy, while that with $m = 0.9$ and $1.0$ exhibits faster entropy decay, demonstrating the RePPO's ability to control the trade-off between exploration and exploitation.

## G  CRAFTAX EXPERIMENT

**Hyperparameters.**   For PPO-V and PPO-V + RND (RND), we used the same hyperparameters as in the original implementation[7]. For RePPO, we adopted the same hyperparameters and modified only the RePPO-specific retry parameter $m$. The full set of hyperparameters is listed in Table 8.

Table 8: Hyperparameters for RePPO, PPO-V and PPO-Q.

| Name (symbol) | Value |
| --- | --- |
| total timesteps | $1.0 \times 10^9$ (1B) |
| learning rate | $2 \times 10^{-4}$ |
| rollout length | 64 |
| parallel envs | 1024 |
| update epochs | 4 |
| minibatch size | 8192 |
| discount $\gamma$ | 0.99 |
| GAE $\lambda$ | 0.8 |
| clip $\varepsilon$ | 0.2 |
| entropy coefficient | 0.0 |
| value loss coefficient | 0.5 |
| max grad norm | 1.0 |
| **RND reelated** | |
| learning rate | $3.0 \times 10^{-4}$ |
| bonus coefficient | 1.0 |
| **RePPO reelated** | |
| RePPO draws $m$ | (1.2, 1.4) |

---

[7]https://github.com/MichaelTMatthews/Craftax_Baselines

## G.1 RESULTS

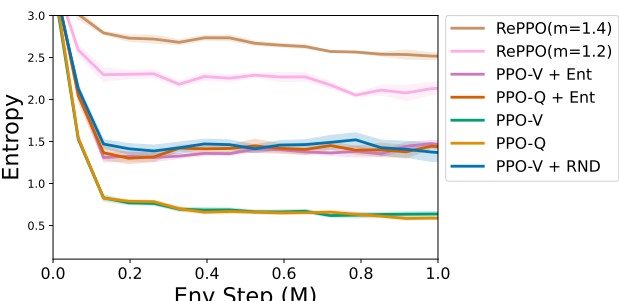

Figure 17: Entropy (Craftax).

We report the policy entropy during training in Fig. 17. As expected, RePPO with $m = 1.2$ and $m = 1.4$ (without an entropy bonus) maintains higher entropy than PPO-V and PPO-V + RND, even when those methods use an entropy bonus. Combined with RePPO's stronger performance compared to PPO without an entropy bonus, this indicates that RePPO effectively promotes exploration without requiring entropy regularization.

## G.2 SPEED BENCHMARK

We also compare the training speed of RePPO with PPO-V and PPO-V + RND in Fig. 18. Computation time is averaged over 5 random seeds. RePPO performs comparably to PPO-V and PPO-Q, and runs faster than PPO-V + RND—an expected outcome, as RePPO does not require additional models beyond those used in PPO-V and PPO-Q.

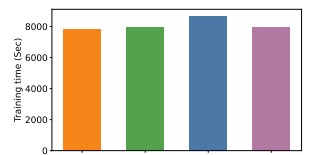

Figure 18: Average training time (Craftax).

## H LLM USAGE

We made limited, assistive use of Large Language Models (LLMs) for presentation-related tasks. In particular, we used LLMs to revise wording for readability, provide minor assistance when organizing proof steps, suggest code refactoring options, and propose small figure improvements. LLMs were not used for research ideation, study design, or the development of substantive scientific contributions.

