# OpenReview forum: "Emergence of Exploration in Policy Gradient Reinforcement Learning via Retrying"
_ICLR.cc/2026/Conference — Submitted to ICLR 2026_

### Official Review · Reviewer_vuQR · 2025-10-29

**Soundness:** 2
**Presentation:** 1
**Contribution:** 1
**Rating:** 2
**Confidence:** 4

**Summary:**

This paper proposes ReMax objective to train explorative reinforcement learning policy. The authors claimed this objective can serve an alternative approach to exhibit exploratory without adding curiosity bonus.

**Strengths:**

1. The objective proposed in this paper, ReMax, is new to the RL community.
2. The authors provides proof to justfiy the objective of the proposed algorithm.

**Weaknesses:**

1. The algorithm is not well-motivated, and the paper is not well-organized.
* The authors should better address the question: "Why not use curiosity bonus? Why should anyone consider using ReMax objective in their work?"
* It is good to have a warm-up section, however, without enough motivation or background, it looks more like a part of methodology section.

2. The experiment result is not convincing, with the following flaws:
* Only 4 environments were used to test RePPO on a small-scale domain like MinAtar. This is too limited.
* The authors claim that ReMax is an alternative exploration method. The experiments are not conducted on the environments that require exploration.
* The baseline should include curiosity-based exploration (explicit bonus). Entropy regularized exploration (used by original PPO paper) has been observed to be insufficient for exploration.

**Questions:**

See weakness. Please try to address the problems mentioned there.

---

> ### Author Response · Authors · 2025-11-21
>
> Thank you very much for the review.
>
> Some main additions we did to address the concerns are: (1) we ran a comparison with the **RND curiosity bonus** method (Sec. 5.1 and Sec. 5.2), (2) we ran experiments on **larger Craftax (Sec 5.2) and Atari  (App. F) environments**, (3) we added further results on the bandit tasks using a softmax policy (Figure 4), the optimal policy under an entropy bonus, to further explain that ReMax empirically yields sublinear regret, whereas softmax has worse regret (linear regret), so it is not a principled exploration strategy. We respond to the questions and concerns below.
>
> ## About Weakness 1 (motivation compared with curiosity bonuses)
> >The authors should better address the question: "Why not use curiosity bonus? Why should anyone consider using ReMax objective in their work?"
>
> To further convey the motivation of the algorithm, we have expanded the introduction section with a paragraph that discusses the extra overhead and complexity of bonus based approaches (written in blue in the updated paper: line 57-61).
>
> Regarding the comment, we would first like to clarify that the main comparison method in the current work is the entropy bonus, not curiosity bonuses in general. We believe that showing an advantage compared to the entropy bonus should be the main criterion for evaluating our work. Curiosity bonuses are primarily intended for “deep” exploration while an entropy bonus changes the learning dynamics and prevents premature convergence of the policy.  In principle, we expect that our method could also contribute to deep exploration, since ReMax has been empirically shown to achieve sublinear regret in bandit problems (Figure 4) that motivated most deep exploration methods in RL (count-based bonuses, and sampling-based methods). However, it may require more improvements such as subtle handling of the uncertainty distribution and it is out of our scope.
> In the current presentation the main benefit is similar to an entropy bonus in the sense that it changes the learning dynamics and maintains a higher policy entropy. We do not claim that curiosity bonuses should not be used; indeed, our method is not mutually exclusive to curiosity bonuses, and both could be used together. We intend for the current work to lay out the main principles for ReMax-based exploration, but foresee that many improvements could be added.
>
> Two more points regarding this: (1) curiosity bonuses like RND require learning a curiosity prediction neural network, whereas our method requires only a parameter $m$. This makes it more similar in complexity to an entropy bonus.
> (2) Regarding entropy bonus approaches, these methods only take into account for the expected value of the returns (not the spread), hence they are not principled exploration methods, and do not yield sublinear regret (see the updated Figure 4). Our ReMax method, on the other hand, empirically gives sublinear regret on the bandit tasks (Figure 4) and adapts to reward uncertainty (Figure 2). These properties make it more appealing compared to an entropy bonus based approach. Moreover, empirically, the performance of ReMax is very strong on MinAtar and other benchmarks (explained further in our response).

---

> ### Author Response · Authors · 2025-11-21
>
> ## About Weakness 2
> > Only 4 environments were used to test RePPO on a small-scale domain like MinAtar. This is too limited.
>
> **To address the concerns, we ran experiments also on Atari and Craftax (https://arxiv.org/abs/2402.16801)** (explained below).
>
> We would also like to first emphasize that **our results on MinAtar are very strong and outperform all previous published results in this domain.** For example, PQN appears to have the previous SOTA results on this benchmark (https://arxiv.org/pdf/2407.04811v2) Figure 8). Our tuned PQN outperforms their results, while RePPO further outperforms this tuned PQN in terms of mean. In particular, on Breakout, we achieve a score of about **175** for RePPO (Figure 8 in our paper), whereas the PQN paper only reports around **75** for their method. Another recent example that used the MinAtar benchmark is the following RLC outstanding paper award winner: (https://openreview.net/pdf?id=LZAafvwVMa)
> They report scores of ~25, ~12, ~45, ~110 for Asterix, Breakout, Freeway and SpaceInvaders respectively for 5M steps. Our corresponding scores at 5M steps are at roughly:
> ~30, ~50, ~60, ~200 and for 10M steps our results improve to ~40, ~175, ~60, ~600. The “Revisiting Rainbow” paper (https://arxiv.org/pdf/2011.14826) that evaluated Rainbow DQN on MinAtar also reports weak results compared to ours. While MinAtar may seem like a simple benchmark, our algorithm does outperform existing competitive implementations on this benchmark.
>
> To further evaluate the exploration capabilities of our method, RePPO, in larger environments, we conducted experiments on both **Craftax and Atari.**
>
> **Craftax** is an open-ended RL environment that requires agents to plan over long horizons and continuously adapt to newly discovered parts of the environment ((https://arxiv.org/abs/2402.16801). **We found that RePPO, without any exploration bonuses, achieves performance comparable to PPO with entropy bonuses or curiosity-based bonuses (RND https://arxiv.org/abs/1810.12894)**. Please refer to Table 1. for the results. Since PPO without bonuses performs poorly, suggesting the need for an exploration mechanism, this result highlights RePPO’s effectiveness in promoting exploration even in large-scale environments. More details can be found in Sec. 5.2.
>
> **To further validate the scalability of RePPO, we ran the Atari experiments** (App. F). We selected 10 environments categorized as hard-exploration problems with dense rewards in (https://arxiv.org/abs/1606.01868). We found that PPO without entropy and RePPO with small retry parameter m=(1, 0.9) already perform strongly, suggesting that these particular tasks do not strongly require stochastic exploration (Figure 14: A). Nevertheless, even in this regime, RePPO with $m>1$ maintains higher entropy than PPO, without bonuses (Figure 16). Overall, these results show that RePPO promotes exploration in Atari while remaining flexible: when little exploration is needed, using a smaller retry parameter (e.g., m = 0.9 or 1.0) still yields competitive performance.
>
> Those results validate the scalability and flexibility of RePPO, by showing it consistently promotes exploration with $m>1$ and achieves high performance when the environment requires exploration (Craftax), while it also can encourage exploitation otherwise with $m \leq 1$ (Atari).
>
> ## About Weakness 3
> >The authors claim that ReMax is an alternative exploration method. The experiments are not conducted on the environments that require exploration.
>
> To address the concern, **we added Craftax, which is designed to gauge exploration ability** (refer to the abstract:  https://arxiv.org/abs/2402.16801). In our earlier responses we also mention that RePPO achieves comparable performance to PPO with entropy and curiosity (RND) bonuses, which validate RePPO's exploratory feature in environments requiring exploration.
>
> Regarding the original MinAtar environments, we partially agree with your critique, e.g., some environments like FreeWay may not require much deep exploration as the agent just has to move up. However, we do not fully agree, as the results clearly show that an entropy bonus significantly improves performance. Moreover, on Breakout, our score with RePPO improves by a lot compared to the previous SOTA (from ~75 to ~175), so exploration may have been a key factor in this result.

---

> ### Author Response · Authors · 2025-11-21
>
> ## About Weakness 4
> >The baseline should include curiosity-based exploration (explicit bonus). Entropy regularized exploration (used by original PPO paper) has been observed to be insufficient for exploration.
>
> **To address this concern, we ran a comparison with RND in MinAtar and Craftax**.
>
> In both benchmarks, we incorporated RND into PPO and ran experiments with and without entropy bonuses. In the MinAtar experiments (Sec. 5.1), **RePPO outperformed RND regardless of the presence of entropy bonuses**, and in Craftax (Sec. 5.2), RND with entropy bonuses performed comparably to our method, alongside PPO with entropy bonuses. Notably, in both settings, **RND exhibits performance degradation when the entropy bonus is removed**, highlighting that stochastic exploration is crucial as a complementary component for curiosity-based methods. Furthermore, in our experiments, RND does not provide a clear performance improvement beyond what entropy bonuses already offer (note that this result is consistent with the results in the original Craftax paper). Importantly, our method achieves comparable, or superior, performance without relying on entropy bonuses, demonstrating its strong exploratory capability across environments with different scales and difficulties.
>
> This question perhaps represents the discrepancy between our intention with the paper and the reviewers request. While entropy regularization may not be enough, it often helps in promoting better learning dynamics and stochastic exploration. We are not currently claiming to propose a method that completely solves the exploration issue in RL. We are proposing an alternative to entropy-based exploration. Curiosity bonuses are another exploration method, and they are not mutually exclusive to our method. From this point of view, we saw entropy bonuses as the natural comparison method, and curiosity bonuses as non-crucial. We hope the reviewer could also see our point of view and consider the benefit that our method shows compared to an entropy bonus based method.
>
> Thank you again for the review, and do not hesitate to ask if there are any remaining questions or concerns.

---

### Official Review · Reviewer_PEMp · 2025-10-30

**Soundness:** 3
**Presentation:** 3
**Contribution:** 2
**Rating:** 4
**Confidence:** 3

**Summary:**

The paper introduces ReMax, a novel RL objective that induces exploration as an emergent property of maximizing the expected maximum return over M retries under uncertainty, without explicit bonuses; it formalizes this in bandits where stochastic policies become optimal for M≥2 and adapts to reward variance, then extends to MDPs via Q-function retries, deriving an estimation-friendly policy gradient using expected improvement and generalizing M to continuous m>0. The proposed RePPO, optimizes ReMax with implicit Q-uncertainty, achieving superior performance and higher entropy than entropy-regularized PPO on experiments.

**Strengths:**

The article exhibits logical coherence by systematically framing exploration as reward maximization under retrying uncertainty, introducing the ReMax objective, demonstrating its efficacy in inducing adaptive stochastic policies in bandits, and extending it to RL via Q-function emulation and an on-policy actor-critic adaptation in RePPO, which delivers robust superior performance on MinAtar with higher scores and policy entropy than entropy-regularized PPO; this is underpinned by a rigorous theoretical foundation, including a closed-form inner expectation and an intriguing expected improvement-based policy gradient derivation that enables single-trajectory estimation and continuous retry parameterization for precise exploration control.

**Weaknesses:**

The reliance on Q-function approximation to emulate retries in non-resettable simulators, a practical workaround for RL extensions that lacks dedicated empirical verification of its approximation quality or potential biases, potentially undermining the objective's fidelity; furthermore, experimental validation is confined solely to the lightweight MinAtar benchmark, limiting evidence of RePPO's scalability and robustness across more complex, diverse, or real-world RL environments, thus insufficiently substantiating the method's broader effectiveness.

**Questions:**

1. Can you provide more validation for the Q-function approximation?For example, comparisons with ground-truth Q values and comparisons with other Q-estimation methods.
2. Can you design more detailed experimental verification? Such as horizontal comparisons of more complex scenarios (Robomimic, D4RL) and different exploration methods (UCB, Thompson, Ornstein, Entropy-regularized...) under various RL algorithms (PPO, TD3...).ation methods.
3. As a higher-order statistical measure, does the variance of EI need further verification?

---

> ### Author Response · Authors · 2025-11-21
>
> Thank you very much for the review.
> ## About Weakness related to the limited experiments
> >furthermore, experimental validation is confined solely to the lightweight MinAtar benchmark, limiting evidence of RePPO's scalability and robustness across more complex, diverse, or real-world RL environments, thus insufficiently substantiating the method's broader effectiveness.
>
> **To address the concerns, we ran experiments also on Atari and Craftax (https://arxiv.org/abs/2402.16801)** (explained below).
>
> We would also like to first emphasize that **our results on MinAtar are very strong and outperform all previous published results in this domain.** For example, PQN appears to have the previous SOTA results on this benchmark (https://arxiv.org/pdf/2407.04811v2) Figure 8). Our tuned PQN outperforms their results, while RePPO further outperforms this tuned PQN in terms of mean. In particular, on Breakout, we achieve a score of about **175** for RePPO (Figure 8 in our paper), whereas the PQN paper only reports around **75** for their method. Another recent example that used the MinAtar benchmark is the following RLC outstanding paper award winner: (https://openreview.net/pdf?id=LZAafvwVMa)
> They report scores of ~25, ~12, ~45, ~110 for Asterix, Breakout, Freeway and SpaceInvaders respectively for 5M steps. Our corresponding scores at 5M steps are at roughly:
> ~30, ~50, ~60, ~200 and for 10M steps our results improve to ~40, ~175, ~60, ~600. The “Revisiting Rainbow” paper (https://arxiv.org/pdf/2011.14826) that evaluated Rainbow DQN on MinAtar also reports weak results compared to ours. While MinAtar may seem like a simple benchmark, our algorithm does outperform existing competitive implementations on this benchmark.
>
> To further evaluate the exploration capabilities of our method, RePPO, in larger environments, we conducted experiments on both **Craftax and Atari.**
>
> **Craftax** is an open-ended RL environment that requires agents to plan over long horizons and continuously adapt to newly discovered parts of the environment ((https://arxiv.org/abs/2402.16801). **We found that RePPO, without any exploration bonuses, achieves performance comparable to PPO with entropy bonuses or curiosity-based bonuses (RND https://arxiv.org/abs/1810.12894)**. Please refer to Table 1. for the results. Since PPO without bonuses performs poorly, suggesting the need for an exploration mechanism, this result highlights RePPO’s effectiveness in promoting exploration even in large-scale environments. More details can be found in Sec. 5.2.
>
> **To further validate the scalability of RePPO, we ran the Atari experiments** (App. F). We selected 10 environments categorized as hard-exploration problems with dense rewards in (https://arxiv.org/abs/1606.01868). We found that PPO without entropy and RePPO with small retry parameter m=(1, 0.9) already perform strongly, suggesting that these particular tasks do not strongly require stochastic exploration (Figure 14: A). Nevertheless, even in this regime, RePPO with $m>1$ maintains higher entropy than PPO, without bonuses (Figure 16). Overall, these results show that RePPO promotes exploration in Atari while remaining flexible: when little exploration is needed, using a smaller retry parameter (e.g., m = 0.9 or 1.0) still yields competitive performance.
>
> Those results validate the scalability and flexibility of RePPO, by showing it consistently promotes exploration with $m>1$ and achieves high performance when the environment requires exploration (Craftax), while it also can encourage exploitation otherwise with $m \leq 1$ (Atari).

---

> ### Author Response · Authors · 2025-11-21
>
> ## About Question 1 (also mentioned in the Weakness)
> >Can you provide more validation for the Q-function approximation? For example, comparisons with ground-truth Q values and comparisons with other Q-estimation methods.
>
> In practice it is difficult to compare with the ground-truth as this is not available. However, we believe the estimation is sufficiently accurate because our PPO-Q implementation is competitive to PPO-V baseline (original PPO), and our results on MinAtar are stronger than any previously published results (see response to previous question). For debugging we also have stored metrics of the value loss (the td error loss), and these have reasonable magnitudes.
>
> ## About Question 2
> >Can you design more detailed experimental verification? Such as horizontal comparisons of more complex scenarios (Robomimic, D4RL) and different exploration methods (UCB, Thompson, Ornstein, Entropy-regularized...) under various RL algorithms (PPO, TD3...).ation methods.
>
> We added more experiments as in our response to the first point in this rebuttal. Regarding Robomimic and D4RL, these are domains with continuous actions whereas we are working with discrete actions at the moment. Our current work has comparisons with PPO with and without entropy, PPO with RND (https://arxiv.org/abs/1810.12894) and PQN (a SOTA DQN-based RL algorithm). TD3 is an algorithm for continuous domains, so we do not compare with it.
>
> ## About Question 3
> >As a higher-order statistical measure, does the variance of EI need further verification?
>
> Out of interest, we conducted an additional comparison and have included the results in the appendix (see Fig. 13), where we see that the variance of EI for $m \ge 1$ is a bit smaller than the typical RL advantage. As a clarifying note, we also follow a standard implementation practice in which the advantages are normalized to have mean 0 and standard deviation 1 when computing the policy update.
>
> Thank you again for the review, and do not hesitate to ask if anything remained unclear or if you have further questions.

---

> > ### Comment · Reviewer_PEMp · 2025-11-22
> >
> > The author has done their best to explain my questions. Considering the work completed so far, I would like to increase my score to 6.

---

> > > ### Author Response · Authors · 2025-11-25
> > >
> > > Thank you very much for your positive review and the time you took to provide such detailed feedback, helping us to enhance the quality of our paper.

---

### Official Review · Reviewer_XGDA · 2025-10-31

**Soundness:** 3
**Presentation:** 3
**Contribution:** 3
**Rating:** 6
**Confidence:** 3

**Summary:**

This paper proposes a novel RL objective, ReMax, which aims to emergently drive exploration by maximizing the expected maximum return over $M$ "retries," without explicit bonuses. To make this practical for on-policy RL, the authors derive a new, estimable policy gradient based on "Expected Improvement" (EI) that works with single-trajectory returns. They also generalize the discrete retry count $M$ to a continuous parameter $m > 0$ for fine-grained control. Their resulting algorithm, RePPO, is shown to outperform PPO on MinAtar and maintain higher policy entropy, demonstrating its exploration capabilities.

**Strengths:**

- The core idea of framing exploration as an emergent property of a retry-based reward objective is elegant and new.

- The derivation of the "Expected Improvement" (EI) policy gradient (Proposition 2) is a solid and clever technical contribution that makes the ReMax objective practical for policy gradient methods.

- Generalizing the discrete retry count $M$ to a continuous parameter $m$ is a useful feature, allowing for fine-grained control over the exploration-exploitation trade-off, as validated in the experiments.

- The empirical results on MinAtar are convincing. RePPO not only outperforms PPO baselines in terms of score but also maintains significantly higher policy entropy without an explicit bonus, strongly supporting the paper's main claim.

**Weaknesses:**

- The paper's motivation relies on (epistemic) uncertainty, but the final algorithm uses the "implicit" non-stationarity of a single Q-network as a proxy. The justification for why this is a sufficient or valid source of uncertainty is weak.

- The method requires sorting Q-values, incurring an $O(K \log K)$ computational cost. This limits the algorithm to discrete action spaces and prevents scaling to continuous control or domains with very large $K$.

- Experiments are only on MinAtar. It is unclear if the performance gains and exploration properties will generalize to more complex environments like the full Atari suite or Procgen.

- As a paper on exploration, it lacks experimental comparisons to other modern exploration methods (e.g., RND, count-based methods), making it hard to gauge its effectiveness relative to the SOTA.

**Questions:**

- If the Q-function converges (becomes stationary), will ReMax stop exploring? This seems to contradict the motivation of exploring under epistemic uncertainty, which should persist in unseen states.

- The ablation shows Q-replacement is critical. Is this technique necessary because the critic is undertrained? Could this issue be solved with more critic updates or a target network, rather than "replacing" the Q-value?

- What is the probabilistic or physical interpretation of a non-integer retry parameter, such as $m=1.2$? Or is it best understood simply as a new hyperparameter to tune exploration intensity?

- Could you please quantify the $O(K \log K)$ overhead more clearly? How much of a bottleneck does this sorting step become as $K$ (the number of actions) grows large?

---

> ### Author Response · Authors · 2025-11-21
>
> Thank you very much for the review.
>
> ## About Weakness 1
> >The paper's motivation relies on (epistemic) uncertainty, but the final algorithm uses the "implicit" non-stationarity of a single Q-network as a proxy. The justification for why this is a sufficient or valid source of uncertainty is weak.
>
> This is an insightful comment and also one of our main concerns hinted at in the conclusions where we mention more explicitly modeling the uncertainty as a future direction. While there is literature on this topic (e.g., “The Phenomenon of Policy Churn”, [https://arxiv.org/pdf/2206.00730](https://arxiv.org/pdf/2206.00730) that shows that the greedy policy over Q-values changes very rapidly), we are also not fully sure that the current approach is valid or the best way. However, our experimental results clearly show that our approach modulates entropy; moreover, the “ReMax with deterministic rewards” paragraph explains that even with no sampling uncertainty, we may expect higher entropy due a slowing of the convergence to the top action. We actually believe that the performance could potentially be improved with a better way to sample from a Q distribution posterior; however, while we have tried several methods, a precise good method has remained elusive to us. We believe this is a good topic for future work, and think it would be difficult to incorporate within the current paper. Our aim with the current work is to lay out the basics, and we believe there are many ways to expand on the current work.
>
> ## About Weakness 2
> >Experiments are only on MinAtar. It is unclear if the performance gains and exploration properties will generalize to more complex environments like the full Atari suite or Procgen.
>
> **To address the concerns, we ran experiments also on Atari and Craftax (https://arxiv.org/abs/2402.16801)** (explained below).
>
> We would also like to first emphasize that **our results on MinAtar are very strong and outperform all previous published results in this domain.** For example, PQN appears to have the previous SOTA results on this benchmark (https://arxiv.org/pdf/2407.04811v2) Figure 8). Our tuned PQN outperforms their results, while RePPO further outperforms this tuned PQN in terms of mean. In particular, on Breakout, we achieve a score of about **175** for RePPO (Figure 8 in our paper), whereas the PQN paper only reports around **75** for their method. Another recent example that used the MinAtar benchmark is the following RLC outstanding paper award winner: (https://openreview.net/pdf?id=LZAafvwVMa)
> They report scores of ~25, ~12, ~45, ~110 for Asterix, Breakout, Freeway and SpaceInvaders respectively for 5M steps. Our corresponding scores at 5M steps are at roughly:
> ~30, ~50, ~60, ~200 and for 10M steps our results improve to ~40, ~175, ~60, ~600. The “Revisiting Rainbow” paper (https://arxiv.org/pdf/2011.14826) that evaluated Rainbow DQN on MinAtar also reports weak results compared to ours. While MinAtar may seem like a simple benchmark, our algorithm does outperform existing competitive implementations on this benchmark.
>
> To further evaluate the exploration capabilities of our method, RePPO, in larger environments, we conducted experiments on both **Craftax and Atari.**
>
> **Craftax** is an open-ended RL environment that requires agents to plan over long horizons and continuously adapt to newly discovered parts of the environment ((https://arxiv.org/abs/2402.16801). **We found that RePPO, without any exploration bonuses, achieves performance comparable to PPO with entropy bonuses or curiosity-based bonuses (RND https://arxiv.org/abs/1810.12894)**. Please refer to Table 1. for the results. Since PPO without bonuses performs poorly, suggesting the need for an exploration mechanism, this result highlights RePPO’s effectiveness in promoting exploration even in large-scale environments. More details can be found in Sec. 5.2.
>
> **To further validate the scalability of RePPO, we ran the Atari experiments** (App. F). We selected 10 environments categorized as hard-exploration problems with dense rewards in (https://arxiv.org/abs/1606.01868). We found that PPO without entropy and RePPO with small retry parameter m=(1, 0.9) already perform strongly, suggesting that these particular tasks do not strongly require stochastic exploration (Figure 14: A). Nevertheless, even in this regime, RePPO with $m>1$ maintains higher entropy than PPO, without bonuses (Figure 16). Overall, these results show that RePPO promotes exploration in Atari while remaining flexible: when little exploration is needed, using a smaller retry parameter (e.g., m = 0.9 or 1.0) still yields competitive performance.
>
> Those results validate the scalability and flexibility of RePPO, by showing it consistently promotes exploration with $m>1$ and achieves high performance when the environment requires exploration (Craftax), while it also can encourage exploitation otherwise with $m \leq 1$ (Atari).

---

> ### Author Response · Authors · 2025-11-21
>
> ## About Weakness 3
> >As a paper on exploration, it lacks experimental comparisons to other modern exploration methods (e.g., RND, count-based methods), making it hard to gauge its effectiveness relative to the SOTA.
>
> **To address this concern, we ran a comparison with RND in MinAtar and Craftax**.
>
> In both benchmarks, we incorporated RND into PPO and ran experiments with and without entropy bonuses. **In the MinAtar experiments (Sec. 5.1), RePPO outperformed RND regardless of the presence of entropy bonuses, and in Craftax, RND with entropy bonuses performed comparably to our method, alongside PPO with entropy bonuses**. Notably, in both settings, RND exhibits performance degradation when the entropy bonus is removed, highlighting that stochastic exploration is crucial as a complementary component for curiosity-based methods. Furthermore, in our experiments, RND does not provide a clear performance improvement beyond what entropy bonuses already offer. Importantly, our method achieves comparable, or superior, performance without relying on entropy bonuses, demonstrating its strong exploratory capability across environments with different scales and difficulties.
>
> ## About Question 1
> >If the Q-function converges (becomes stationary), will ReMax stop exploring? This seems to contradict the motivation of exploring under epistemic uncertainty, which should persist in unseen states.
>
> The main idea is that as long as we keep visiting unseen states, then we will be sampling new Q-values, and the exploration will continue. Once there are no more unseen states and the Q-function converges, then the method is expected to stop exploring and converge to the greedy policy. We believe this is a desirable behavior, as exploration is needed while the method has still not fully explored the environment. In practice however, the “damping effect” seen in Sec. 2.1, paragraph named “ReMax with deterministic rewards” section also slows convergence and promotes maintaining a higher entropy for longer.
>
> ## About Question 2
> >The ablation shows Q-replacement is critical. Is this technique necessary because the critic is undertrained? Could this issue be solved with more critic updates or a target network, rather than "replacing" the Q-value?
>
> Our motivation for Q-replacement as written in the paper (line 408-412) is that the expected improvement of an action over itself should be 0, so by using the replacement technique, we can enforce this. In practice, even with excellent critic training, the EI over the action itself will not be 0 without using the replacement technique. We believe part of the reason for the criticality of this approach is the damping effect that ReMax has on the convergence to the top action. With replacement, the total EI will tend to 0 as we converge on the top action, but without replacement, because the EI of the action over itself may not be 0, the damping effect may be reduced. The critic is fitted adequately, as evidenced by the strong comparable performance of our PPO-Q baseline compared to the original PPO with V critic.
>
> ## About Question 3
> >What is the probabilistic or physical interpretation of a non-integer retry parameter, such as $m$? Or is it best understood simply as a new hyperparameter to tune exploration intensity?
>
> It is the natural extension of the number of tries to a continuous parameter by simply replacing the integer $M$ with a real value $m$ in the equation. It is unclear how to physically interpret this in finite terms.
>
> ## About Question 4 (also mentioned in Weakness)
> >Could you please quantify the overhead more clearly? How much of a bottleneck does this sorting step become as (the number of actions) grows large?
>
> The runtime results in Figure 12 (MinAtar) and Figure 18 (Craftax) show that the computational overhead of RePPO is very small.
> In MinAtar, the difference in computation time is only about 5–10%, and in Craftax the difference between PPO and RePPO is less than 1%. Since Craftax requires substantially more computation per step due to the larger network size and complicated environment logic, this suggests that the additional sorting operation in RePPO is negligible compared to the cost of policy training and environment interaction.
> This is also consistent with how fast modern CPUs can sort small or medium-sized arrays: even an array with 10,000 elements can typically be sorted in under 1 ms. In practice, the sorting operations introduced by RePPO add almost no runtime overhead.
>
> Thank you again for the review, and do not hesitate to ask if you have any remaining concerns or questions.

---

> > ### Comment · Reviewer_XGDA · 2025-11-25
> >
> > Thanks for the response, most of my concerns are well addressed, I will increase my score

---

> > > ### Author Response · Authors · 2025-11-25
> > >
> > > We appreciate your positive evaluation of our paper. Your insightful feedback has been instrumental in improving the quality of our work.

---

### Official Review · Reviewer_ERuW · 2025-10-31

**Soundness:** 3
**Presentation:** 3
**Contribution:** 3
**Rating:** 6
**Confidence:** 2

**Summary:**

The main idea of this paper is to propose an exploration strategy based on the idea to "bias" the classical RL objective by taking the best outcome over M trials (the ReMax objective).
The article discusses this approach in both a multi-armed bandit framework and RL framework.
In the RL framework, an algorithm RePPO is proposed: a combination of ReMax with PPO.
The paper proposes both several theoretical results regarding the elaboration of some closed-forms (proofs are given in appendix), as well as some empirical evaluations.

**Strengths:**

Exploration is an ever-lasting question in RL and bandits. The main advantage of the approach explored in this paper is its simplicity in terms of additional parameters: it only requires an integer M.
Even if I did not have the time to perform an in depth reading of the paper, I enjoyed going through it. The presentation is somehow original, with some didactical elements such as questions encouraging the reader to think proactively while reading.

**Weaknesses:**

Some notations are considered as implicitly defined, such as $\Delta^{K-1}$. What are precisely per-action values in a bandit problem, expected values or random variables associated with arm pulling ?

**Questions:**

I may clearly have missed some information, but my main question would be related with the possibility that optimizing policies according to ReMax may induce some sort of bias ?
The article speaks much more about exploration than exploitation, whereas these two notion are usually intertwined. Is it because, in practice, the authors did not observed any over-exploitation behaviors (because of too-low values for M) ?

---

> ### Author Response · Authors · 2025-11-21
>
> Thank you very much for the review.
>
> > Even if I did not have the time to perform an in depth reading of the paper, I enjoyed going through it.
>
> We are happy to hear that you enjoyed reading the paper, and would encourage you to go through it in detail with the extra time in the review process to form a confident opinion. One pointer regarding what to check is that the current review mentions an *integer* $M$, but we later extended our method to a *real valued* $m$ that aids in tuning, and can also induce exploitation behavior. We respond to your questions and other comments below.
>
>
> ## About Question 1
> > I may clearly have missed some information, but my main question would be related with the possibility that optimizing policies according to ReMax may induce some sort of bias?
>
> The aim of ReMax is to modify the learning dynamics to enhance exploration during training, but ultimately converge to the optimal policy. In deterministic environments, the optimal policy remains unchanged between the ReMax objective and the classical RL objective
>
> $\pi^* = \pi^*_\textup{ReMax}$. This is because in this setting, the optimal action is fixed, and even under $M$ retries, picking the top action with probability 1 is optimal. The same logic applies to our RL algorithm in stochastic settings, because the evaluation is based on the Q-value, which is a deterministic fixed value after convergence.
>
> However, when there is stochasticity (for example via implicit posterior sampling during training due to distribution shift), then exploratory stochastic policies become optimal under the ReMax objective (Figure 6: B). Moreover, even with no stochasticity, ReMax modifies the gradients so that they become smaller as the probability converges on the optimal action; thus, enhancing exploration by slowing down premature convergence (See Sec. 2.1, paragraph named **ReMax with deterministic rewards.**).
>
> In summary, ReMax adds an **“exploratory bias” during training**, but **in ideal settings would still converge to the optimal policy for the RL objective** (i.e., in tabular settings, we expect this, but in practice policy capacity affects the result).
>
> We find it useful to also contrast this with an entropy bonus: an entropy bonus adds a clear bias, so that the softmax policy becomes optimal, and note that this never converges to the optimal RL policy (unlike our method).
>
> ## About Question 2
> > The article speaks much more about exploration than exploitation, whereas these two notion are usually intertwined. Is it because, in practice, the authors did not observed any over-exploitation behaviors (because of too-low values for M) ?
>
> **Our method allows controlling the exploration-exploitation behavior.** Note that we extended our method to work for continuous parameters $m>0$, and we find that for $m\in(0, 1)$, we can induce greater exploitation than the standard RL objective. This is illustrated in **Fig. 3**, where a low $m<1$ value causes the objective function to become sharp and the gradient to accelerate toward the optimal action, and in **Fig. 6B**, we can also see that for $m=0.9$, the entropy reduces faster than for $m=1$ (which corresponds to the standard RL objective). Thus our method can not only promote exploration, but can also promote exploitation. However, in practice, good performance requires exploration, so we discuss this more in our paper.
>
> ## About Weakness 1
> >Some notations are considered as implicitly defined, such as $\Delta^{K-1}$. What are precisely per-action values in a bandit problem, expected values or random variables associated with arm pulling ?
>
> Thank you, $\Delta$ is the probability simplex. We have clarified the notations in line 43-44.
>
> Regarding the bandit problem, there are two notions here: the values used for training the ReMax policy, and the values obtained when actually pulling the arm. The value obtained from pulling the arm is a random variable $\mu + \epsilon$, where $\mu$ is the true mean of the arm, and $\epsilon$ is an observation noise. This follows the standard bandit setup. However, for training the ReMax policy, we sample $\mu$ from the current estimated posterior $\mu_i \sim \Pi_t$, and do not add any observation noise. This happens “in simulation” using the current posterior estimate to optimize the ReMax policy. Then we sample an action from the ReMax policy, and pull this arm on the real bandit problem. For clarity, we note that such posterior estimates $\Pi_t$ are also used in Thompson sampling and UCB algorithms. In Thompson sampling, one samples from the posterior, and picks the top arm from the sample, in UCB, one picks the arm based on upper confidence intervals of the posterior. ReMax uses this posterior in a different way to obtain the sampling policy.
>
> Thank you again for the review, and do not hesitate to ask if you have any remaining concerns or questions.

---

### Public Comment · ~Haque_Ishfaq1 · 2025-11-17
**Possible related work and baselines that models Q-function uncertainty**

Hi Authors,

We would like to point out few related works that might of interest for this work. In [1], [2], and [3], we propose Thompson sampling based algorithms where we also model the Q-function uncertainty. Especially in [2] and [3] we propose Langevin Monte Carlo based DQN algorithms that samples from the approximate posterior of the Q-function. We also perform experiments in 8 hard atari games. Our code is publicly available in case you want to use it as baseline:

(i) https://github.com/panxulab/LSVI-ASE
(ii) https://github.com/hmishfaq/LMC-LSVI

Finally we would appreciate if you consider discussing these works in your Extended Related Work section where you discuss posterior sampling.

Thanks!


[1] Ishfaq et al., Randomized Exploration for Reinforcement Learning with General Value Function Approximation, ICML 2021

[2] Ishfaq et al., Provable and Practical: Efficient Exploration in Reinforcement Learning via Langevin Monte Carlo, ICLR 2024

[3] Ishfaq et al., More Efficient Randomized Exploration for Reinforcement Learning via Approximate Sampling, RLC 2024

---

> ### Author Response · Authors · 2025-11-21
>
> Thank you for your interest in our paper, for pointing out these related works, and for sharing the code. We agree that [1–3] are relevant, since Thompson-sampling–based methods are promising for explicitly modeling Q-function uncertainty. We have added a discussion of these papers in the extended related work section.
> In the new version of our draft, we added Atari experiments for the larger-scale update, and we now have three environments in common with [2, 3] (Alien, Hero, and Qbert). We did not include the methods as baselines mainly because our PPO baseline outperforms them (on Hero and Qbert) or is similar (on Alien), while the total number of frames is 4/5 of [2, 3] (see Figure 15).
> Although explicit posterior modeling is beyond the scope of our current work, we think it is an important direction for future research, and we consider your methods promising candidates to combine with ours. In particular, Langevin Monte Carlo sampling of Q-values from an uncertainty distribution [2, 3] seems very well suited to ReMax.
> Again, thank you very much for introducing these relevant works.

---

### Author Response · Authors · 2025-11-21
**General Response**

**We sincerely appreciate all the helpful feedback provided by the reviewers**.

We have updated our paper, and highlighted the changes in blue.

The reviewers acknowledged the novelty of our objective, ReMax, maximizing return over multiple retries under epistemic uncertainty, as well as the usefulness of our proposed algorithm, RePPO, especially its flexible control of the retry parameter enabled by our new policy gradient formulation.

The main concerns were related to the experimental validation:
(1) the absence of large-scale environments such as Atari, and
(2) the lack of comparisons with curiosity-based exploration methods.

We addressed these concerns by adding additional experiments:

**(1) Large-scale environments: Craftax and Atari**.
Craftax (https://arxiv.org/abs/2402.16801) is an open-ended RL benchmark that requires exploration and adaptation to newly encountered situations. In both Craftax (Sec. 5.2) and Atari (App. F), RePPO with a retry parameter greater than 1, without any exploration bonuses, naturally promotes exploration, validated by the maintained entropy over the course of training. In Craftax in particular, RePPO significantly outperforms PPO with no entropy bonus, and achieves performance comparable to PPO with entropy or curiosity bonuses. We believe those results helps to address the first concern by demonstrating effective exploration in large-scale environments.

**(2) Curiosity-based baseline (RND)**.
We added RND as a baseline on MinAtar and Craftax. In MinAtar (Sec. 5.1), RePPO outperformed RND regardless of whether entropy bonuses were used. In Craftax, RND (https://arxiv.org/abs/1810.12894) with entropy bonuses performed comparably to PPO with entropy bonuses and to our method. In both environments, RND showed clear performance degradation when entropy bonuses were removed, further highlighting that RePPO encourages exploration effectively even without entropy bonuses.

As a final note, we would like to emphasize the main motivation of our paper: to provide a simple, principled alternative to entropy-based exploration by showing that **ReMax can promote exploration without explicit bonuses, especially entropy**. We believe our experiments validate this basic premise. At the same time, there are many promising directions for future work, including explicit uncertainty modeling and combining ReMax with modern curiosity-based bonuses.

---

> ### Comment · Area_Chair_cUEr · 2025-11-21
> **Author-Reviewer Discussion**
>
> Dear reviewers,
>
> Please review the authors' response and adjust your rating accordingly. If you have any further questions, please discuss with the authors further.
>
> AC

---

### Author Response · Authors · 2025-12-03
**Summary of Rebuttal and Discussion**

We would like to express our sincere gratitude to the Area Chair and the reviewers for their time and constructive feedback. While we previously summarized our rebuttal in the General Response, given the recent change in Area Chairs following the incident, we are providing a consolidated summary here to facilitate your evaluation.

During the discussion phase, we received responses from 2/4 reviewers before the incident on November 27 (XGDA, PEMp) who reacted positively to our responses (please check the discussion for details). The remaining two reviewers were unfortunately not able to respond before the discussion was disabled.

**Strengths and Key Highlights**
- The reviewers acknowledged the novelty of our ReMax objective and the practical utility of the RePPO algorithm, particularly its flexible control of exploration via the retry parameter.
- Additionally, during the discussion, we highlighted our significant performance gains on the MinAtar benchmark. Specifically, we showed that RePPO substantially outperforms previous state-of-the-art methods (e.g., achieving ~175 on Breakout vs. the previous best of ~75), further validating the effectiveness of our approach.

**Main concerns**

The primary concern regarding the limited scale of experiments was addressed by conducting two additional sets of evaluations:
- Large-Scale Experiments (Craftax & Atari): Three reviewers expressed concern regarding the limited experimental scope of our original paper. In response, we conducted additional experiments on Craftax (https://arxiv.org/abs/2402.16801) (Sec. 5.2) and Atari (App. F). These results confirm that RePPO scales effectively and promotes exploration in complex environments without relying on explicit bonuses.
- Comparison with Curiosity Baselines (RND): Reviewers XGDA and vuQR requested comparisons to curiosity-based bonuses in addition to entropy bonuses. We discussed the motivational differences between ReMax and curiosity bonuses, clarifying that they are not mutually exclusive (see response to Weakness 4 for Reviewer vuQR). Furthermore, to fully address their concerns, we added comparisons with RND (https://arxiv.org/abs/1810.12894) in MinAtar (Sec. 5.1) and Craftax (Sec. 5.2). RePPO performs comparably to or better than RND without requiring auxiliary networks.

**Detailed Summary of Revisions**

Below is a comprehensive summary of the revisions and discussions addressed during the rebuttal. We abbreviate "Weakness" as W and "Question" as Q. We refer to Reviewers ERuW, XGDA, PEMp, and vuQR as R1, R2, R3, and R4, respectively.

| Issue | Corresponding review | Revision / Response |
| :--- | :--- | :--- |
| **Scale of Experiments** | W2(R2), W(R3), W2(R4) | We added experiments on Craftax and Atari (Sec. 5.2, App. F). |
| **Curiosity Baselines** | W3(R2), W4(R4) | We added RND comparisons. RePPO matches or exceeds RND performance, serving as a robust alternative to entropy bonuses without needing auxiliary networks (Sec. 5.1 and Sec. 5.2). |
| **Bias of ReMax** | Q1(R1) | Answered in the official comments and added a sentence in Sec. 2.1. |
| **Exploration property** | Q2(R1), Q1(R2) | Answered in the official comments. |
| **Clarification of bandit** | W1(R1) | Answered in the official comments. |
| **Uncertainty estimation** | W1(R2) | Answered in the official comments. |
| **Validity of Q-replace** | Q2(R2) | Answered in the official comments. |
| **Interpretation of $m$** | Q3(R2) | Answered in the official comments. |
| **Computation overhead** | Q4(R2) | Added speed benchmark on App. E for Craftax. |
| **Q function estimation** | Q1(R3) | Answered in the official comments. |
| **Variance of EI** | Q3(R3) | Added the report of the variance of EI-based advantage in Fig. 13. |
| **Motivation of ReMax** | W1(R4) | Discussed in the official comment. Also, we added a paragraph in introduction and updated the bandit experiment (Fig. 4). |
| **Environment validity** | W2(R4) | Added Craftax experiment in Sec. 5.2. |


We have uploaded the revised paper reflecting these changes (highlighted in blue). We are confident that we could complete the answer to the main research question “Can we promote exploration without adding explicit bonuses by optimizing ReMax in RL?”(posed in the introduction line 70-72) leveraging the help of instructive comments by all the reviewers.

---

### Meta-Review · Area_Chair_45RV · 2026-01-06

**Summary:**

This paper proposes ReMax, a retry-based reinforcement learning objective that induces stochastic exploration by maximizing the expected best outcome over multiple draws, together with an EI-based policy gradient and a PPO instantiation (RePPO). While the objective and derivations are technically sound and the empirical results are carefully executed, I do not believe the paper meets the conceptual novelty bar for acceptance. At a high level, ReMax does not demonstrate a clear conceptual advantage over existing local exploration mechanisms based on stochasticity, most notably entropy regularization and its many annealed or adaptive variants. Empirically, the paper primarily shows that ReMax maintains higher policy entropy for longer, but does not convincingly establish that it explores meaningfully different or better states than entropy-based methods, nor that it addresses exploration challenges that cannot already be handled by standard entropy scheduling or related stochastic regularizers. Moreover, the core objective is mathematically equivalent to best-of-N / pass@K optimization, a line of work that has already been explored in both RL-adjacent and policy-optimization settings. While the authors extend this idea with an EI-based gradient estimator and a continuous retry parameter, these contributions are incremental and do not fundamentally distinguish ReMax from prior work on optimizing pass@K or max@K objectives. As a result, the paper is best viewed as a careful re-derivation and engineering of existing retry-based objectives within PPO, rather than a new exploration principle. Given the lack of a clear conceptual or algorithmic advance beyond existing stochastic exploration methods and prior best-of-N optimization work, I recommend rejection.

**Reviewer Scores:**

NA

---

### Decision · Program_Chairs · 2026-01-26

Reject